# Cutaneous Melanoma: An Overview of Physiological and Therapeutic Aspects and Biotechnological Use of Serine Protease Inhibitors

**DOI:** 10.3390/molecules29163891

**Published:** 2024-08-16

**Authors:** Ana Paula De Araújo Boleti, Ana Cristina Jacobowski, Tamaeh Monteiro-Alfredo, Ana Paula Ramos Pereira, Maria Luiza Vilela Oliva, Durvanei Augusto Maria, Maria Lígia Rodrigues Macedo

**Affiliations:** 1Laboratory of Protein Purification and Their Biological Functions, Food Technology and Public Health Unit, Federal University of Mato Grosso do Sul (UFMS), Campo Grande 79070-900, MS, Brazil; apboleti@gmail.com (A.P.D.A.B.); anacristinaj@gmail.com (A.C.J.); tamaehamonteiro@hotmail.com (T.M.-A.); aninha12357@gmail.com (A.P.R.P.); 2Departamento de Bioquímica, Universidade Federal de São Paulo (UNIFESP), São Paulo 04023-062, SP, Brazil; mlvoliva@unifesp.br; 3Divisão de Ciências Fisiológicas e Químicas, Serviço de Bioquímica, Instituto Butantan, São Paulo 05585-000, SP, Brazil; durvanei.maria@butantan.gov.br; 4Department of Pharmaceutical Sciences, Food, and Nutrition, Federal University of Mato Grosso do Sul, Campo Grande 79070-900, MS, Brazil

**Keywords:** skin cancer, etiology, staging, targeted therapy, proteolytic activity

## Abstract

Background: Metastatic melanoma stands out as the most lethal form of skin cancer because of its high propensity to spread and its remarkable resistance to treatment methods. Methods: In this review article, we address the incidence of melanoma worldwide and its staging phases. We thoroughly investigate the different melanomas and their associated risk factors. In addition, we underscore the principal therapeutic goals and pharmacological methods that are currently used in the treatment of melanoma. Results: The implementation of targeted therapies has contributed to improving the approach to patients. However, because of the emergence of resistance early in treatment, overall survival and progression-free periods continue to be limited. Conclusions: We provide new insights into plant serine protease inhibitor therapeutics, supporting high-throughput drug screening soon, and seeking a complementary approach to explain crucial mechanisms associated with melanoma.

## 1. Introduction

With a high probability of metastasis, melanoma is recognized as the most common and aggressive malignant neoplasia among skin tumors [1,2]. Melanoma develops when melanocytes, which produce pigment, grow abnormally in the basal layer of the epidermis. Melanocytes have melanosomes with the following two types of pigments: pheomelanin (yellow/red) and eumelanin (brown/black). Eumelanin is the primary photoprotective pigment in the skin, while pheomelanin generates reactive oxygen species under UV radiation, leading to DNA damage [3]. 

Melanogenesis can be stimulated in a paracrine, autocrine, or intracrine manner. The melanocyte-stimulating binding hormone (α-MSH) promotes the production of eumelanin by binding to the Melanocortin 1 Receptor (CM1R), which then triggers an increase in Intracellular Cyclic Adenosine Monophosphate (cAMP) levels and activates the cAMP-response-element-binding protein (CREB). Along with SOX10 (Sex-Determining Region Y-box 10), and PAX3 (Paired Box 3) transcription factors, this leads to activation of MITF (Microphthalmia-associated Transcription Factor), resulting in increased expression of crucial enzymes for melanogenesis from amino acid tryptophan [3].

Conversely, melanocytes are derived from the Neural Crest (NC), a group of versatile precursor cells between the neural tube and the ectoderm. This region is also responsible for generating neurons, glial cells, and various other cell types. During early embryogenesis, neuroectodermal cells form the neural plate, which folds and closes to form the neural tube. NC cells migrate, in a delamination process, along the dorsolateral route to the epidermis, where they differentiate into melanocytes in the basal layer. A melanocyte subpopulation may originate from cells that migrated in the ventral route, proliferating in the dermis before attaching to the epidermis [4]. 

The current classification of melanoma is based on the foundational contributions of CMGovern and Clark [5,6]. Most cutaneous melanomas originate from epidermal melanocytes and progress through two stages as follows: initial radial growth (RPG) and subsequent vertical growth. In the radial growth phase, lesions appear as circular spots, while in the vertical growth phase, a tumor forms and grows both below and above the skin. The initial stage, RGP, exhibits the ABCDE signs, where “A” stands for asymmetry, “B” for irregular border, “C” for color variation, “D” for diameter (>4 mm initially), and “E” for evolution. Minor injuries can be diagnosed, but strict criteria prevent the “overdiagnosis” of harmless injuries [7,8,9].

When evaluating suspected pigmented lesions, they observe changes in new skin lesions, considering pigmentary variations, asymmetry, ill-defined edges, ulcer formation, nodules in pre-existing lesion development, and pigment loss as signs of possible melanoma [10]. Definitive diagnosis requires biopsy and histopathological analysis to distinguish melanoma from other conditions [9].

Melanoma neoplasms have an aggressive metastatic potential, with a high propensity to spread, especially to the lymph nodes, brain, and lungs [11]. Metastases occur in organs such as the pancreas, bones, small intestine, and adrenal glands [12]. Surgical excision is the main therapeutic approach for early melanoma, while adjuvant therapies, chemotherapy, immunotherapy, and the use of vaccines are complementary options for tumors in advanced stages [13].

The prognosis for metastatic melanoma is generally unfavorable, with an estimated 10% survival rate extending beyond 10 years [14]. Distant metastases have a significantly worse prognosis compared with local spread. Treatment options for this type of cancer involve surgical excision, which is the primary therapeutic approach for early melanoma. Regarding advanced-stage tumors, additional treatment options such as chemotherapy, immunotherapy, and vaccines can be used alongside each other [9]. 

Recent studies have explored several potential targets for drug development. Protease inhibitors (PIs) of plant origin are particularly notable, emerging as promising candidates with extensive applications in biotechnology and medicine. Understanding protein interactions relies heavily on these substances, which also serve as important tools in developing new compounds to control diseases and treat various ailments [15,16,17]. In this review, we discuss melanoma biology. In addition, we explore the different subtypes of melanoma and detail the staging process used to determine the cancer’s location and extent. We also review current treatment options for melanoma and investigate the anticancer effects of serine protease inhibitors across various melanoma cell lines.

## 2. Epidemiology

Previously considered a rare disease before World War II, cutaneous melanoma (CM) has seen its global incidence rise over 60-fold since then. The earliest recorded cases were in Connecticut in 1935, with rates under 5 cases per 100,000 people annually, during a time of notably low melanoma rates [18]. By 2020, 325,000 cases were diagnosed, primarily in individuals over 50, with 57,000 deaths reported that year [2].

CM incidence rates worldwide depend on numerous factors, including exposure to solar radiation and ultraviolet rays (UVRs), access to health care (prevention and treatment), and individual genetic factors. Among these inducing factors, approximately 75% are attributed to UVRs [19]. Associated risk factors include higher smoking prevalence, alcohol consumption, unhealthy eating, obesity, and metabolic diseases. Family history is one of the most critical risk factors for melanomas [20], where genetic mutations of the same standard can be transmitted to generations. 

The incidence of cutaneous melanoma (CM) is higher in white-skinned populations, who are more vulnerable to the effects of sunscreens [21,22]. In contrast, African and Asian populations have better protection because of the pigmentation of their skin. They are more likely to develop acral melanomas, which occur on the palms, soles of the feet, or under the nails of the hands or feet, as well as mucosal melanomas, but at lower rates [23]. 

For all genders, there is a significant increase in incidence in the age group of 50 years or older. Although mortality rates and trends have declined in recent decades, the global incidence has risen, particularly among older age groups (over 50) and men [24]. Among the older adult population (over 65), the incidence of CM is higher, but it is not uncommon among young adults under 30 [2]. Throughout the world, CM is more prevalent in adult males, with approximately 174,000 new cases reported each year, compared with 151,000 new cases in females. This trend is reversed in specific areas like East and West Africa, Northern Europe, and Melanesia. The prevalence of CM is higher in males over the age of 75, while females in the same age group are more likely to experience invasive melanoma [22]. 

Gender differences can be observed in the anatomical location of the CM, with men more commonly experiencing it in the trunk and women in the lower limbs [25]. The ears, face, neck, and scalp are commonly affected by CM in older individuals, regardless of gender [26]. 

The highest annual incidence rates are in Australia and New Zealand (42 men and 31 women in 100,000 inhabitants), followed by Europe (19 men and 19 women in 100,000 inhabitants), North America (18 men and 14 women in 100,000 inhabitants), and North European (17 men and 18 women in 100,000 inhabitants). Africa and Asia have the lowest annual incidence rates (less than one person in 100,000 inhabitants) [27]. It is observable that the occurrence in specific countries, including Australia and the United States, has reached a steady state, possibly because of a robust campaign advocating for lifestyle modifications and the adoption of UVR protectors, besides racial mixing [23]. 

Brazil is a continental country with a tropical climate and a high incidence of UVRs [28]. Among those at higher risk are individuals engaged in professional activities that require exposure to intense solar radiation, such as tourism, fishing, agriculture, and construction. The population has a rich historical composition, which includes European, indigenous, and African American ancestry, being an important factor. Brazilian CM incidence and mortality patterns vary across geographical regions because of the country’s population diversity, regional differences, socioeconomic factors, demographic factors, and ethnic factors. The highest rates have been reported in the southern region [29]. 

In 2015, melanoma incidence rates in Brazil were around 3.5 new cases per 100,000 people for men and 3.3 for women. In terms of mortality rates, there were 0.92 deaths per 100,000 men and 0.71 deaths per 100,000 women during the same period [30]. Data from the National Cancer Institute (2020) showed that CM incidence rates rose to 4.03 new cases per 100,000 inhabitants for men and 3.94 new cases per 100,000 inhabitants for women [31].

## 3. Melanoma Subtypes

Over the years, melanoma has been classified into several categories based on the tissue from which the primary tumor originates. These classifications also consider the etiological association with sun exposure or lack thereof, determined by specific mutations, anatomical location, and epidemiological characteristics. Melanomas that form on sun-exposed skin are subdivided based on the histopathological degree of cumulative solar damage (CSD) in the surrounding skin. They are categorized as low or high CSD, depending on the presence of associated solar elastosis [8]. 

Melanomas with low CSD include cases of superficial spreading melanoma, while those with high CSD encompass lentigo maligna and desmoplastic melanomas. The “non-solar” category includes acral melanomas, melanomas in congenital nevi, blue nevi, Spitz melanomas, and mucosal melanomas [8,32].

### 3.1. Lentigo Maligna 

Over 75% of all cases of melanoma in situ (MIS) are attributed to lentigo maligna (LM), which is the primary subtype. This subtype is characterized by the abnormal growth of melanocytes along the skin’s junction, specifically in sun-exposed areas like the head and neck, as well as infiltrating hair follicles and sweat ducts. It is most commonly diagnosed in individuals aged 65 to 80 [33,34]. As a form of “in situ” melanoma, LM’s malignant cells are confined to the epidermis. Although it grows slowly, LM can progress to dermal invasion, becoming lentigo maligna melanoma (LMM) [33,34,35,36,37]. Hyperpigmentation and the formation of lentigines around melanoma scars are common symptoms of LM/LMM, which primarily develop in areas exposed to the sun. Poorly defined and irregularly shaped patches [35,36,37,38] may be observed. 

Genetic changes in neurofibromin 1 (NF1), B-Raf proto-oncogene (BRAF)V600K, non-V600E (glutamic acid substitution) mutations, neuroblastoma RAS viral oncogene homolog (NRAS), and, to a lesser extent, KIT proto-oncogene receptor tyrosine kinase (KIT) are among the varying mutations in lentigo maligna melanoma (LMM) compared with superficial spreading melanoma (SSM) [39]. The NF1 gene makes individuals more susceptible to multiple tumors, particularly those that arise from the neural crest. This autosomal dominant disorder has a high mutation rate, with approximately 50% being new mutations. Neurofibromin, the protein encoded by the NF1 gene, plays a crucial role in negatively regulating the Ras/MAPK pathway, which enhances cell proliferation and suppresses apoptosis. A deficiency in neurofibromin is linked to the development of various tumors, both benign and malignant. Individuals with NF1 are at a higher risk of developing optic pathway glioma, glioblastoma, Malignant Peripheral Nerve Sheath Tumor (MPNST), breast cancer, sarcoma, leukemia, Gastrointestinal Stromal Tumor (GIST), pheochromocytoma, duodenal carcinoid tumor, and melanoma [40].

The mitogen-activated protein kinase (MAPK) pathway, responsible for transmitting extracellular signals to the nucleus and regulating cell proliferation and differentiation, is frequently dysregulated in cutaneous melanoma [39]. Approximately 90% of cutaneous melanomas exhibit abnormal activation of the mitogen-activated protein kinase (MAPK) pathway, primarily because of mutations in the BRAF gene. Somatic mutations in BRAF are prevalent in 37% to 60% of melanomas, most commonly occurring at position 600. About 80% of these mutations are of the V600E type, involving a substitution of valine with glutamic acid, while 5% to 12% are V600K mutations, where lysine substitutes valine. The protein kinase encoded by the BRAF gene regulates the phosphorylation of mitogen-activated protein kinase kinase (MEK) and maintains the protein in an inactive state through hydrophobic [40,41]. Variations in genetic mutations give rise to diverse treatment choices for metastatic cutaneous melanoma. This type of melanoma has a high number of mutations, particularly those caused by UV exposure, which makes it more responsive to immunotherapy using checkpoint inhibitors [42].

### 3.2. Desmoplastic Melanoma 

Desmoplastic melanoma (DM) is a rare histological type of melanoma. It was initially described in 1971 by Conley and colleagues, who reported seven cases of imperceptible superficial melanotic lesions in the head and neck regions. These lesions progressed into an aggressive, infiltrating tumor with metastasis potential [42]. Histologically, it is characterized by spindle cells surrounded by thickened desmoplastic stroma and abundant collagen [42,43,44,45,46]. Morphologically, it is identified by a paucicellular dermal tumor with an irregular, poorly defined contour and a low-to-moderate density of melanocytes [47]. The specific cause of DM is not known, but it is commonly found in areas that have prolonged exposure to ultraviolet (UV) radiation. A typical manifestation is a hardened lesion on sun-damaged skin, exhibiting characteristics of UV mutations [42,43,48]. 

Desmoplastic melanoma has a distinct genetic profile compared with other melanoma subtypes. Mutations in the B-Raf proto-oncogene (BRAF) and neuroblastoma RAS viral oncogene homolog (NRAS), common in other melanomas, are less frequent in this variant. However, mutations in the neurofibromin 1 (NF1) gene are more prevalent, leading to the activation of the RAS-mitogen-activated protein kinase (MAPK) pathway, which promotes cell proliferation. Additionally, alterations in the tumor protein p53 (TP53) gene, resulting in the loss of p53 tumor suppressor function, are commonly observed, contributing to genomic instability and tumor progression [49,50].

By inactivating NF1, the activation of the mitogen-activated protein (MAP) kinase pathway is prolonged, enhancing cellular proliferation [51]. Common mutations in BRAF and neuroblastoma RAS viral oncogene homolog (NRAS), typically present in other melanomas, are absent in this context. However, genetic alterations have been identified in NF1, casitas B-lineage lymphoma (CBL), erb-b2 receptor tyrosine kinase 2 (ERBB2), MAP kinase kinase 1 (MAP2K1), MAP kinase kinase kinase 1 (MAP3K1), BRAF, epidermal growth factor receptor (EGFR), protein tyrosine phosphatase non-receptor type 11 (PTPN11), MET proto-oncogene (MET), Ras-related C3 botulinum toxin substrate 1 (RAC1), SOS Ras/Rac guanine nucleotide exchange factor 2 (SOS2), NRAS, and phosphatidylinositol-4,5-bisphosphate 3-kinase catalytic subunit alpha (PIK3CA). Some of these are potential candidates for targeted therapies [43]. Treatment options include surgical excision, sentinel lymph node biopsy, adjuvant radiotherapy to the tumor bed, and systemic chemotherapy. Recently, DM has been observed to be associated with multiple genetic mutations and has shown responsiveness to targeted therapies [47,48].

### 3.3. Acral Melanoma

Malignant melanomas arise from melanocytes in the basal layer of the skin, mucosal epithelium, and uveal tract. As melanocytes are distributed throughout the body, various clinical types of melanomas, such as acral melanoma (AM), are diagnosed [52,53]. AM is a relatively unique tumor within the melanoma spectrum [53,54]. It is found on glabrous (hairless) skin, affecting sun-protected sites like the palmar, plantar, and subungual surfaces [52,54,55].

Unlike other forms of cutaneous melanoma, acral melanoma is not influenced by risk factors like UV ray exposure, fair skin type, family or personal history of melanoma, or pre-existing melanocytic nevi [53,54,56]. This type of melanoma rarely presents mutations in the BRAF and NRAS genes, common in other melanomas, but has a high frequency of mutations in the KIT gene, which can activate the MAPK signaling pathway. In addition, acral melanoma frequently exhibits complex structural alterations, including amplifications and deletions of chromosomes, particularly affecting the cyclin D1 (CCND1), cyclin-dependent kinase 4 (CDK4), and telomerase reverse transcriptase (TERT) genes [56]. 

There is oncogenic activation of the CDK4 pathway through copy number gains in CDK4 and CCND1 and loss of cyclin-dependent kinase inhibitor 2A (CDKN2A). Melanoblasts with CCND1 amplification were detected in the secretory portion of eccrine glands among human melanoma cells, suggesting that these cells could give rise to acral melanomas. CCND1 encodes cyclin D1, a protein essential for the G1/S transition as it forms a complex with CDK4 or CDK6. This cellular distribution may explain the characteristic parallel ridge pattern of acral melanoma [57]. The diagnosis of AM is difficult, being commonly confused with other dermatological conditions, which ends up leading to diagnoses at a more advanced stage [57] and greater development of metastasis [54]. 

Currently, the combination therapy of BRAF and MEK inhibitors is being used to treat BRAF-mutated melanoma, resulting in swift but only partial responses in many cases [53]. Nivolumab and pembrolizumab, which are anti-PD-1 antibodies, are widely used worldwide to treat patients with advanced melanoma. Recent studies in various cancer immune checkpoint inhibitors show that tumors with higher mutational burdens often respond well to anti-PD-1 antibodies [52].

### 3.4. Spitz Melanoma

In previous times, spitzoid melanomas were classified according to their cytomorphological attributes. The analysis of genomic data has demonstrated that a considerable number of these cases possess genomic features resembling those of low cumulative sun damage (low-CSD) melanomas, frequently displaying B-Raf proto-oncogene serine/threonine kinase (BRAFV600E) mutations. The World Health Organization (WHO) updated its classification, defining Spitz melanoma (SM) as the cancerous form of Spitz nevi (SNs), which can be distinguished by their morphology and genomic characteristics [58]. Genetic fusions involving anaplastic lymphoma kinase (ALK), neurotrophic receptor tyrosine kinase 1 (NTRK1), ROS proto-oncogene 1 (ROS1), rearranged during transfection (RET), and B-Raf proto-oncogene serine/threonine kinase (BRAF) genes are frequently observed in this type of melanoma. These fusions lead to the continuous activation of signaling pathways, such as the mitogen-activated protein kinase (MAPK) pathway, which support cell growth and survival. Mutations in the Harvey rat sarcoma viral oncogene homolog (HRAS) gene are also observed, contributing to the activation of the phosphatidylinositol 3-kinase (PI3K)-AKT pathway. These fusions and mutations lead to unique biological behavior, with the potential for rapid growth but a lower metastasis rate when compared with other melanomas. Understanding these molecular features is crucial for developing targeted and personalized treatments for patients with Spitz melanoma [58,59,60].

Atypical Spitz tumors (ASTs) are a type of melanocytic neoplasm that exhibits intermediate features between benign Spitz nevi and malignant melanomas. Spitz nevi (SNs) occur more frequently in childhood, while atypical Spitz tumors (ASTs) and Spitz melanomas (SMs) are probably more common in older ages, although conclusive data on risk factors are unknown [61,62]. They are characterized by their unusual clinical and histopathological features, making their diagnosis challenging. ASTs often present as pigmented or non-pigmented skin lesions, typically appearing as solitary, or multiple papules or nodules, and/or epithelioid melanocytes. In Spitz nevi (SNs), these cells exhibit amphophilic hyaline cytoplasm, large nuclei with regular membranes, pale chromatin, and prominent nucleoli. SNs frequently feature a junctional component with cell nests, cleft artifacts, and globoid eosinophilic “Kamino bodies” at the interface. These cells extend into the papillary dermis, often reaching the reticular dermis, with a tendency to mature toward the base [63,64]. 

A subset of spitzoid lesions, classified in the low degree of cumulative solar damage (low-CSD) pathway, is associated with loss of expression of the tumor suppressor BRCA1-associated protein 1 (BAP1) and expresses BRAFV600E protein [64]. These predominantly intradermal tumors exhibit cytological variability, ranging from minimal atypia to large epithelioid cells with striking atypical features such as nuclear pleomorphism and increased mitotic activity [58].

### 3.5. Mucosal Melanoma 

Mucosal melanoma (MM) is a melanoma that arises on mucous membranes, predominantly in genital, oral, nasal, and conjunctival areas, with rare occurrences in other mucous membranes. The occurrence of this form of melanoma is similar in frequency among individuals of all races and represents a significant portion of melanomas in regions where white people are not considered high-risk. The factors that contribute to the risk are unknown, with no clear connections to sun exposure, chemical carcinogens, or viral involvement [65].

Mucosal melanomas can progress through an early stage (radial growth phase—RGP) with characteristics similar to “ABCDE”, making them clinically recognizable in visible areas like the vulva, oral cavity, and conjunctiva. However, when in nasal sinuses or visceral organs, they are rarely detected at this early stage [66]. Because visualizing can be difficult, mucosal melanomas frequently appear as sizable tumors that invade and harm nearby tissues. These symptoms may come with bleeding, pain, or discomfort. Recent studies reveal a molecular resemblance between vulvar and vaginal melanomas, suggesting a close relationship. Despite this, there is a difference between vulvar lesions, which typically involve the skin, and mucosal melanoma, which occurs outside the realm of gynecology [66,67].

Recent genetic sequencing studies have identified alterations in neuroblastoma RAS viral oncogene homolog (NRAS), BRAF, neurofibromin 1 (NF1), KIT, splicing factor 3b subunit 1 (SF3B1), tumor protein p53 (TP53), and sprouty-related EVH1 domain-containing protein 1 (SPRED1), revealing potential targeted therapeutic strategies for MM. Patients suffering from this condition display a reliance on the MAPK pathway, although their mutation patterns vary from those seen in cutaneous melanoma (CM). NRAS and BRAF mutations are less common in MM, while SF3B1 mutations and KIT alterations are more frequent. Given the lower prevalence of targetable BRAF mutations in MM, it is necessary to validate targets in other alterations within the MAPK pathway [66].

### 3.6. Melanoma Arising in a Congenital Nevus

Congenital nevi, present at birth in approximately 1% of newborns, resemble acquired nevi and fall into the following three categories: giant (covering large areas and typically not removable), intermediate (suitable for surgical excision), and small (under 2.5 cm in diameter). Melanomas in giant congenital nevi often develop in childhood, with a lifetime occurrence rate ranging from 1% to 30% [8,67].

Melanomas can arise from congenital moles in either the junctional or dermal regions. Junctional component lesions, especially those with a radial growth pattern (RGP), may bear similarities to low-CNS (Copy-Number-Signature-low) melanomas. In comparison, individuals who come from the dermal component have noticeable distinguishing features. A growing melanoma can be clinically disguised by the underlying pigmented nevus, often with hair. It is important to differentiate this condition from cellular and proliferative nodules found in congenital nevi. These nodules are typically benign and develop within the first year of life. Sometimes, a biopsy is needed to confirm the diagnosis [8,68,69].

Melanoma arising from a congenital nevus is characterized by activating mutations in the neuroblastoma RAS viral oncogene homolog (NRAS) gene, often involving the loss of the normal allele and amplification of the mutant NRAS. These changes promote MAPK pathway activation, contributing to malignant transformation. Additional mutations, such as changes in TP53 and CDKN2A, are required for tumor progression. Despite the infrequency of BRAF mutations in this melanoma type, genotyping for both NRAS and BRAF is recommended as a result of the effectiveness of available inhibitors. Large congenital nevi are often present with genomic instability, which highlights the importance of monitoring and early intervention [69,70]. In melanomas linked to giant congenital nevi, telomerase reverse transcriptase (TERT) expression can be upregulated through epigenetic modifications, specifically methylation, while the tumors maintain the wild-type genotype [71,72].

## 4. Melanoma Staging Phases

As our knowledge of melanoma’s complex biology grows, staging systems continue to improve. Since the 1990s, significant advancements in managing cutaneous melanoma have emerged by introducing lymphatic mapping and sentinel lymph node biopsy. The American Joint Committee on Cancer’s Cancer Staging Manual (AJCC) periodically reviews the Tumor, Node, Metastasis (TNM) system to ensure it reflects a precise understanding of the disease. The current TNM system for melanoma (Table 1), detailed in the eighth edition of the AJCC, categorizes the disease into stages ranging from I to IV [73,74].

### 4.1. Primary Staging of Melanoma: Stages I and II

Tumor thickness and ulceration are the primary criteria for categorization on the T scale (Figure 1). Primary melanomas are divided into four substages (T1–T4) based on tumor thickness, with further substaging in the presence of ulceration. Melanomas without nodal metastases are classified into five substages (IA/B and IIA/B/C) based on tumor thickness and ulceration [73,75]. Breslow thickness is the most crucial prognostic factor for patients with clinically localized primary cutaneous melanomas. It has also been demonstrated to be the most consistently reproducible factor in melanoma histopathological reporting [76,77].

Tumor thickness, assessed by Breslow thickness, is pivotal for prognosis and management. The AJCC recommends measuring from the top of the granular layer to the deepest invasive cell, to the nearest 0.1 mm. T categories are defined in integers (1.0, 2.0, 4.0 mm), with a significant cutoff point around 0.7 to 0.8 mm for T1 melanoma [77,78], and “T2” stage melanomas as being between 1.01 and 2.00 mm thick. Thus, measuring Breslow thickness around the 1.00 mm cutoff for T1 staging has particularly important implications for both pathological staging and clinical management [77].

### 4.2. Melanoma Staging for Regional Lymph Nodes: Stage III

The identification of lymph node metastases in patients with melanoma indicates a more aggressive tumor biology and correlates with a decrease in survival [79]. Among patients with nodal metastases (stage III), the clinical status of the lymph nodes (palpable or not) and the number of metastatic lymph nodes are the main prognostic factors [78]. Over the last three decades, the surgical approach to regional lymph node metastases in malignant melanoma patients and our comprehension of prognostic features have significantly evolved. Previously, melanoma lymph node metastases were categorized as “microscopic” or “macroscopic” [80]. In the current staging system, this terminology has been reformulated as “clinically occult” or “apparent”, reflecting how these metastases are identified [81]. 

Patients with clinically hidden nodal disease are categorized as N1a, N2a, or N3a based on the number of lymph nodes involved unless non-nodal locoregional disease is present (microsatellite, satellite, or in-transit metastases). Therefore, N1a denotes clinically hidden lymph node metastasis (detected only by SLN biopsy); N2a indicates two or three clinically hidden (detected by SLN biopsy); and N3a represents four or more clinically hidden (detected by SLN biopsy) [82]. 

Clinically evident nodal disease is classified as N1b, N2b, or N3b based on the number of lymph nodes involved. N1b denotes clinically identified lymph node metastasis; N2b represents two or three nodes, with at least one clinically identified; and N3b indicates four or more nodes, with at least one clinically identified, or the presence of any number of tangled nodes [78,82].

### 4.3. Metastatic Melanoma: Stage IV

Patients with stage IV melanoma typically have a challenging prognosis, with a median survival ranging from 8 to 10 months. To refine prognosis assessment, stage IV melanoma patients are categorized into subgroups as follows: those with isolated skin metastases (IVa), lung metastases (IVb), or metastases to other organs (IVc). These subdivisions have associated 5-year survival rates of 18.8%, 6.7%, and 9.5%, respectively. Patients with cutaneous melanoma metastases (stages IIIb or IVa) often undergo treatment with local excision; however, this approach does not always eradicate microscopic malignant cells in transit, often leading to the development of future tumors [83,84].

For patients with metastatic disease, the anatomical site of metastasis is used to define M categories, which are divided into four subcategories as follows: (a) M1a: skin, soft tissue, including muscle and/or non-regional lymph node; (b) M1b: lung; (c) M1c: visceral sites not belonging to the central nervous system (CNS); and (d) M1d: CNS [77]. The site of metastasis is the most crucial predictor of outcome in melanoma. M1a metastasis indicates a relatively better prognosis compared with other anatomical sites. M1b is considered an “intermediate” prognosis. M1c now includes only non-CNS (central nervous system) viscerais sites in the latest edition and has a worse prognosis than M1b. For patients with CNS metastasis, the M1d category has the poorest prognosis among all M categories [78].

## 5. Therapeutic Strategies and Medications Currently Used

Melanoma treatment is defined according to staging and considers individual factors of the disease and the patient [85], namely, medical history, lesion thickness, growth, metastasis, and cell mutations. In general, the most common treatments are surgery, radiotherapy, chemotherapy, immunotherapy, and conditioning therapy [86]. In addition to these treatments, some clinical studies also involve the prospect of vaccines [85]. The area of melanoma study has developed a lot in recent times; with the most diverse treatments, even the most advanced and metastatic cases have shifted from being less responsive to becoming more responsive. It is important to remember that various treatments can cause side effects and need to be monitored.

### 5.1. Conventional Diagnosis and Therapy

A crucial step in defining clinical management is the diagnosis based on biopsy. According to the American—National Comprehensive Cancer Network [87]—and European—European Society of Medical Oncology (ESMO, 2023)—international guidelines, a narrow excision of 1–3 mm should be performed on suspicious pigmented lesions. The most important prognostic indicator for cutaneous melanoma is Breslow thickness. Sentinel lymph node biopsy (SLNB) [88] can also be performed, which recent studies have linked to a better prognostic determination [89,90]. The overwhelming majority of cases still recommend diagnosis by shave biopsy. Shave biopsy, despite being easy and quick, can result in inaccurate determinations about the patient’s real condition [91] and will soon become obsolete. On the other hand, some studies suggest the adoption of alternatives such as gene expression profiling (CP-GEP, Merlin Assay^®^) [92], the use of emerging biomarkers for accurate prognostic determination of patients, and a more appropriate definition of the therapeutic strategy [93].

#### 5.1.1. Surgical Excision

Surgery is the chosen treatment for most melanoma cases. Despite its high effectiveness, patient conditions often limit this approach, such as possible tissue intolerance, specific co-morbidities, and the patient’s non-acceptance [94]. In cases where surgical excision is utilized, cancerous tissue is removed along with a portion of healthy tissue surrounding the lesion, depending on the disease stage. For melanoma in situ, excisional surgery with removal of a safety margin of healthy tissue is performed. 

Recently, several clinical reports demonstrating the efficacy of Mohs micrographic surgery in melanoma therapy have considered factors such as patient safety for those diagnosed with melanoma in situ (MIS) and invasive melanoma (IM), low treatment costs, tumor staging risks, and potential failure in sentinel lymph node biopsy (SLNB). These studies suggest that Mohs surgery yields results like or even better than local tumor excision [95]. On the other hand, other studies, such as those by Huerta et al. [96], suggest that the current evidence does not support the equivalence of Mohs micrographic surgery to wide excision for cutaneous melanoma. A more detailed, individualized evaluation is always crucial. Additionally, given the complex context of melanoma, recent studies have encouraged combined and systemic approaches to increase the likelihood of successful therapy [97].

#### 5.1.2. Radiotherapy

Melanoma cells are generally radioresistant, as they have low differentiation. This is attributed to cell cycle arrest induced by DNA damage, high proliferative capacity, and an efficient enzyme repair system that mitigates radiation damage when applied at low doses [98]. Radiotherapy is less frequently used in the early stages of melanoma because of its characteristics. It is more commonly adopted when cancer has metastasized, particularly to the central nervous system, as it can be challenging to reach with systemically administered drugs [86].

In the therapeutic context, radiotherapy can be used for distant metastases and subcutaneous recurrences to aid in disease control and symptom relief. This method can be associated with conditions such as hyperthermia and even immunotherapy. Boron neutron capture therapy might be an option alongside standard treatments, although its clinical application is still limited [99]. Radiotherapy has demonstrated variable efficacy in treating local melanoma, particularly in cases of brain metastases. Consequently, there is an increasing encouragement for combined therapies [100]. Recent clinical data and trials have shown improvements with combinations such as intensity-modulated therapy combined with stereotactic surgery and immunotherapy, especially with checkpoint inhibitors (discussed in subsequent sections), which have exhibited synergistic potential [101,102,103]. Despite these promising results, radioinduced resistance and associated adverse effects remain significant challenges that need further investigation and resolution.

#### 5.1.3. Chemotherapy

Chemotherapy involves using drugs, administered orally or by injection, to disrupt the growth and induce the death of cancer cells. Its use depends on the stage and condition of the patient. There are systemic and regional chemotherapy approaches, where drugs are applied directly to the target site, whether in an organ or body cavity, including cerebrospinal fluid. For cancers in the limbs, isolated limb hyperthermic perfusion may be used. A tourniquet is placed on the limb to reduce blood flow, and then a heated solution of medication is injected directly into the area. This technique allows for high doses of medication to be delivered locally [104]. 

The topical application of these chemotherapeutic agents is also an alternative and allows the administration of higher doses with lower toxicity [94]. According to the National Cancer Institute, the following drugs are used for melanoma treatment and have been approved by the Food and Drug Administration (FDA) (Appendix A). Overall, these include interleukins, BRAF oncoprotein suppressors, and checkpoint inhibitors. They are used in various disease stages, including cases that cannot be treated by surgery, or administered after the surgical procedure, to prevent recurrence and in cases of metastasis [85].

### 5.2. Innovative Therapies 

#### 5.2.1. Topical Treatments

Topical treatment is a therapeutic option for patients with subclinical or multiple lesions in a widespread area, where applying a large amount of the active substance with minimal toxicity is necessary. This approach is less feasible with systemic administration [105]. Topical treatment is also considered when surgery is not an option for the patient, or when there are limitations for more invasive procedures because of factors like tissue intolerance or certain comorbidities [94]. The topical dermal route is easy to administer, non-invasive, and minimizes systemic effects, thus enhancing patient compliance. Semisolid dosage forms, such as gels, are particularly advantageous because of their ability to adhere well to the application site, prolong the release of active ingredients, and provide excellent spreadability. Gels also tend to have fewer long-term stability issues compared with other semisolid forms and are relatively straightforward to formulate, accommodating a variety of active compounds [106]. According to Cancer Research UK, topical treatments are applied directly to the lesion site and are generally administered once or twice a day for 3 to 4 weeks, following medical advice [107].

There are several reports in the literature about the potential of the compound 5-fluorouracil (5-FU) as a topical chemotherapeutic agent, used in monotherapy or combined therapy [108,109]. Another topical chemotherapeutic is imiquimod, belonging to the imidazoquinoline family. It is an immunomodulator that acts as a toll-like receptor (TLR) agonist in macrophages, Langerhans cells, dendritic, and antigen-presenting cells. This modulatory function can activate NF-κB and release interleukins, chemokines, and cytokines such as IFNγ, IFNα, and TNF. This immunomodulatory effect has led to the use of imiquimod as an adjuvant in melanoma treatment and in cases where surgery is impossible. Its therapeutic indication is a 5% cream [110,111]. In sequence, ingenol mebutate, an important activator of protein kinase C (PKC), was extracted from the sap of a plant, Euphorbia peplus [94]. In clinical trials, it is believed that the subsequent effect of PKC activation is the promotion of cell death by necrosis through increased calcium influx and swelling of mitochondria [112], as well as promoting vascular damage, mainly through the ERK1/2 pathway [113]. 

Finally, another therapeutic alternative already in use is vitamin A analogs or retinoids. Besides their widespread application in dermatology and cosmetics, retinoids are also crucial in the differentiation process of keratinocytes, impacting this cellular process by interacting with specific nuclear receptors for retinoic acid [94]. Because of this interaction and its immunomodulatory effects, retinol has been widely used in topical therapy, often in pharmacological combinations [114,115].

#### 5.2.2. Targeted Anticancer Therapies: BRAF or MEK Mutation 

Targeted therapy is a cancer treatment that specifically targets proteins and genes responsible for regulating the growth, division, and spread of cancer cells. As our understanding of genetic mutations and protein mechanisms in cancer cells advances, researchers can develop more precise treatments [107]. In melanoma treatment, targeted therapy is particularly revolutionary because it causes less damage to healthy cells. These treatments focus on specific aspects of the immune system or on problematic genes and molecules, such as inhibiting the mitogen-activated protein kinase (MAPK) pathway in tumors with BRAF or MEK mutations [116], or targeting pathways like cyclin-dependent kinases, such as CDK4/6 [117].

The BRAF gene serves as the “switch gene” responsible for regulating skin cell multiplication. Mutations in this gene promote abnormal activation that culminates in uncontrolled cancer growth [109]. The BRAFV600 mutation is noteworthy as 40 to 50% of melanoma cases involve patients with this specific gene mutation, promoting a constitutively activated MAPK pathway [118]. 

Vemurafenib, dabrafenib, and encorafenib are approved BRAF inhibitors (BRAFi) for advanced and metastatic melanoma cases [119]. Vemurafenib is a highly selective inhibitor, and its use in metastatic melanoma treatment has shown positive responses in 50% of patients [120,121]. Despite its anticancer potential, there are cases of renal and hepatic insufficiency in oral and intravenous administration and resistance to treatment [86,121]. Dabrafenib is another FDA-approved BRAFi for V600 mutation cases but is associated with various cases of resistance and therapeutic inefficiency as well [109]. As a therapeutic alternative, combinations of BRAFi with other inhibitors are also used, such as the cyclin-dependent inhibitors (CDK—ribociclib, abemaciclib, and palbociclib), the C-KIT inhibitor (imatinib) [122], and MEK inhibitor (MEKi), like binimetinib, cobimetinib, and trametinib [109]. These have shown greater efficiency than BRAFi, becoming a new therapeutic standard for advanced melanoma with BRAFV600 mutation. BRAFi and MEKi are capsules or pills normally taken once or twice a day [123].

Phase 1 and 2 clinical studies with the BRAF inhibitor vemurafenib showed a 50% response rate in patients diagnosed with the BRAF V600E mutation. Additionally, a phase 3 randomized study comparing vemurafenib with dacarbazine in untreated metastatic melanoma demonstrated that vemurafenib resulted in greater overall survival and reduced disease progression [124]. In the Columbus study conducted by Dummer et al. [125], the combination of the BRAF inhibitor encorafenib and the MEK inhibitor binimetinib was compared with vemurafenib alone. This combination proved to be more effective, showing increased overall survival for patients with the BRAF mutation. Similarly, the phase 3 coBRIM study, published by Ascierto et al. [126], demonstrated the efficacy of combining cobimetinib with vemurafenib for treating patients with advanced melanoma with the V600 mutation. The efficacy of this pharmacological combination was further confirmed in phase 3 randomized studies, COMBI-d [127] and COMBI-v [128], which assessed the quality of life of patients receiving the combined therapy of dabrafenib and trametinib versus vemurafenib. These studies consistently showed that the combination therapy resulted in healthier patients and more effective treatment outcomes. The COMBI-r study, based on daily clinical practice, further confirmed these results [129]. Other inhibitors, like imatinib targeting C-KIT, and multikinase inhibitors, can also assist in treating KIT mutations, a pro-oncogene responsible for encoding the transmembrane tyrosine kinase receptor. This mutation is typically linked to acral and sun-aggravated melanoma. Despite affecting a small percentage of the population, mutations in this gene are crucial for cell survival, proliferation, and differentiation [130,131]. These multikinase inhibitors are represented by drugs like nilotinib, sunitinib, and dasatinib [131,132]. 

Here, we review the main mutations associated with cutaneous melanoma, focusing on the BRAF, MEK, and KIT genes. We provide a detailed comparison of mutation rates and types among patients with cutaneous melanoma, other melanoma subtypes such as acral and mucosal melanoma, and different ethnic groups. Recent studies have shown that persistent tumor mutation burden (pTMB) is directly associated with improved responses to immune checkpoint blockade therapy. Xu et al. [133] developed a melanoma model based on pTMB, which demonstrated a higher incidence of genetic alterations and greater immune cell infiltration in the high-pTMB group. Another study identified the COL4A3 gene as a significant therapeutic target in UV-related cutaneous melanoma, as it inhibits cell proliferation and migration [134]. Li et al. [135] suggested using immunological and transcriptomic profiling to develop diagnostic and prognostic models for cutaneous melanoma, identifying crucial genes involved in immune response and melanoma progression.

Mutations in the BRAF, NRAS, and KIT genes activate various signaling pathways, notably the MAPK/ERK pathway, which can influence cytoskeletal organization. This activation can lead to actin reorganization through modulators like profilin, impacting the migratory and invasive abilities of tumor cells [133]. Constitutive activation of these pathways, such as the BRAFV600E mutation, can elevate the expression and activity of cytoskeletal regulatory proteins, including profilin. Changes in profilin expression can thus modulate tumor cell invasion and metastasis, processes heavily dependent on actin dynamics [136]. Studies have shown that profilin regulation directly affects cellular invasion and proliferation, making profilin and its regulators potential therapeutic targets [134]. For instance, overexpression of COL4A3, a gene identified as significant in melanoma, can influence profilin regulation, actin dynamics, and the invasive capacity of tumor cells [135].

#### 5.2.3. Immunotherapy

Immunotherapy can be performed in patients regardless of the presence of mutations. It uses medicines that stimulate the immune system to destroy cancer cells. Proteins derived from the immune system itself and synthetic molecules are used in this approach [107]. These drugs act on immune checkpoints and directly control the patient’s immune response, helping to maintain the balance of this system [116]. Since the 1990s, immunotherapy has been developing, initially with the FDA approval of interferon-α2b, and later interleukin IL-2 (highly toxic). In the 2000s, the immune checkpoint inhibitor anti-CTLA-4 was approved for advanced and inoperable cases. Subsequently, the anti-PD-1 and, in 2022, the combined therapy of the anti-LAG-3 monoclonal antibody, relatlimab associated with nivolumab, was approved for metastatic cases [137,138].

CTLA-4 is a transmembrane receptor from the immunoglobulin group expressed on activated T lymphocytes and directly acts on the inhibition and activation of T cells [116]. CTLA-4 plays a pivotal role in immune function, particularly in antitumor responses, as it strongly binds to co-activator molecules CD80 and CD86 expressed by monocytes, similar to CD28 on T cells. Its interaction with CD80 and CD86 negatively regulates T-cell activation and inhibits IL-2 release. Conversely, when CTLA-4 binds to B7, it initiates an inhibitory cascade, suppressing T cells. Thus, anti-CTLA-4 antibodies disrupt the binding of CD28 to B7, boosting T cell activation and enhancing immune function against cancer [132]. 

The use of ipilimumab, an anti-CTLA-4 drug, has demonstrated a significant increase in patient survival but has also been associated with several adverse reactions due to the overactivation of T cells. Interestingly, ipilimumab has already been tested in association with a glycoprotein peptide vaccine—gp100 [116], and is also used in combination with dacarbazine, among other options, including new pharmaceutical forms of drug delivery that combine peptides with immunotherapeutics [120,139,140].

Immunotherapy also includes PD-1 blockers; in the clinic, drugs like nivolumab (human antibody) and pembrolizumab (humanized IgG4 monoclonal antibody) are approved [116]. Both drugs target programmed cell death receptor (PD-1), binding to it and subsequently inhibiting its ligands, PD-L1 (CD274) and PD-L2 (CD273), which are expressed in immune and non-immune cells. PD-1 is expressed on natural killer cells, monocytes, activated B and T cells, and PD-L1 and PD-L2 ligands. These ligands are expressed not only on normal cells but also on tumor cells. PD-1 inhibitors negatively regulate the protein responsible for cell proliferation, IL-2 secretion, and T-cell inhibition. This leads to activation effects and enhances the immune system’s recognition, resulting in immune-mediated antitumor effects [141].

In addition to standalone therapies using single inhibitors (monotherapy), pharmacological combinations have shown promise in melanoma treatment. An example of this combination is the concurrent blockade of CTLA-4 and PD-1 with ipilimumab and nivolumab, which has demonstrated a synergistic effect [120,142]. These new therapeutic strategies aim to overcome drug resistance, promoting better treatment conditions, greater efficiency, and longer-lasting effects [132]. 

In a phase 3 study examining the combination of atezolizumab, cobimetinib, and vemurafenib, it was demonstrated that this combination was more effective than the individual drugs, particularly for patients with advanced BRAFV600 mutations [143]. Another phase 2 study investigated the safety of combining a PD-L1 inhibitor (lambrolizumab) with an anti-CTLA-4 inhibitor (ipilimumab). That study showed greater tumor regression in patients who received the combination therapy, even in cases where tumors initially developed when treated only with ipilimumab [144]. Later, the same research group published an article analyzing patient responses to a combination of nivolumab (a PD-1 inhibitor) and ipilimumab (a CTLA-4 inhibitor). This combination therapy resulted in significant improvements in overall survival and progression-free survival compared with individual therapies. Although the treatment led to more grades 3 and 4 adverse effects, these were manageable and allowed for the maintenance of patient quality of life [145]. The data confirm that combined therapy with CTLA-4 and PD-1/PD-L1 inhibitors is effective, enhancing both patient survival and quality of life.

#### 5.2.4. Toll-like Receptor-9 (TLR-9) Agonists 

TLR (Toll-like receptors) belong to a family of receptors expressed in immune cells that recognize molecular patterns of pathogens, known as Pathogen-Associated Molecular Patterns (PAMPSs). Activation of these PAMPS occurs through TLR-9, present on the endoplasmic reticulum and intracellular vesicles, which are then translocated to endosomes, exposing them to PAMPS. TLR-9 not only releases PAMPS but also triggers a downstream intracellular cascade, activating NFkB and secreting IFN-1 [146,147], turning on CD8+ T cells in the tumor microenvironment. This sequence of actions promotes a better antitumor response [148,149]. Because of the potential for TLR-9 to trigger anticancer responses, its agonists are being increasingly tested in the clinic for melanoma and various other types of cancer. These agonists can be administered locally at the lesion or systemically.

Similar to other types of therapeutic strategies, TLR-9 agonists are also being tested in combined therapies, including chemotherapy, immunotherapy, or radiotherapy [116]. An example of this promising association is the case of tilsotolimod, a synthetic TLR-9 agonist administered in combination with ipilimumab [150]. Other examples of TLR-9 agonists include SD-101, which is currently under investigation in combination with pembrolizumab for patients with advanced melanoma [148], and vidutolimod (CMP-001), which has a distinctive structure, resembling an immunogenic film similar to a virus, triggering the production of antibodies.

Recent clinical studies have explored the anti-melanoma potential of treatments involving TLR-9 agonists. Vidutolimab, when combined with pembrolizumab, achieved a 25% response rate in patients with PD-1 refractory melanoma. Additionally, the combination demonstrated a safe profile [151]. Another trial investigated the compound CMP-001, also in combination with pembrolizumab, through a phase 1b study with two parts. This combination was administered intratumorally and showed good tolerance as well as a significant local tumor response, highlighting the therapy’s potential for combined administration [152].

#### 5.2.5. Adoptive Cellular Therapy

The therapeutic strategy involving tumor-infiltrating lymphocytes (TILs) is a form of cellular immunotherapy. This technique entails extracting lymphocytes from the tumor site through surgical excision, potentially combined with lymphodepleting chemotherapy during hospitalization. The extracted lymphocytes are then expanded in vitro using a culture medium with IL-2, tested for reactivity against autologous tumor cells, selecting the most reactive lymphocytes, and finally incubating them with anti-IL2 and anti-CD3 antibodies before infusion back into the patient. In response to the body’s exposure to cells, the body recognizes and attacks the tumor cells. This procedure typically lasts around 6 weeks [153,154,155]. Treatments with TILs have shown better results and increased patient survival compared with cases treated with other targeted therapies, such as ipilimumab [154]. The currently approved medications include pegylated interferon 2-α, interleukin-2, and interferon α-2b [107].

Recently, the FDA approved adoptive cell therapy using tumor-infiltrating lymphocytes (TILs), specifically lifecycle, for the treatment of advanced melanoma. The phase 3 trial, conducted for the first time in 2022, demonstrated a 50% reduction in solid tumors compared with standard therapy [156]. The FDA’s approval of this therapy, along with its growing acceptance among clinicians, represents a significant advancement in improving patient response rates and survival.

#### 5.2.6. Gene Therapy

This therapeutic approach involves inserting viral or bacterial suicide genes into cancer cells. This transfection makes previously non-toxic metabolites toxic to the cell, causing cell death. A prodrug is then used, acting as a bystander through gap junctions, effectively killing all cancer cells, whether they were transfected with the suicide genes or not [157]. This relatively understudied technique is promising because some stem cells naturally have the ability for tumor-tropic migration, making their use as a “cellular vehicle” viable [158].

Several genes can be used in this way, including the cytosine deaminase (CD) gene from Escherichia coli, which converts the non-toxic antifungal agent 5-fluorocytosine (5-FC) into 5-FU (discussed in the topical therapy section), and thymidine kinase gene from the herpes simplex virus (HSV-tk) with ganciclovir (GCV) as a prodrug [159]. The conversion of non-toxic metabolites into toxic ones occurs through the action of kinases, culminating in the inhibition of DNA polymerase, cell cycle delay in the S and G2 phases, mitochondrial damage, and death through both apoptosis and necrosis [86].

Phase 1/2 studies investigated the efficacy of DF6002, an oncolytic agent, in combination with nivolumab for patients with advanced or metastatic melanoma. One study includes elements of suicide gene therapy, aiming to enhance the antitumor response while minimizing side effects [160]. Another study by the Henry Ford Health Institute examined a genetically modified adenovirus incorporating suicide genes. These genes are designed to interfere with DNA replication, promoting cell death, particularly when used in conjunction with chemotherapy or radiotherapy [161]. The therapies discussed in this review are summarized in Figure 2.

This Figure 2 summarizes diagnostic procedures and therapeutic approaches for cancer treatment. It includes different biopsy techniques (Figure 2A), such as incision, excision, shave, and punch biopsy, and the use of sentinel lymph nodes biopsy (SLNB) for detecting cancer spread, with biomarkers as a future diagnostic tool. (Figure 2B) Conventional therapies depicted are surgery for removing cancerous tissues, chemotherapy administered orally, topically, or via injection to disrupt cell growth and induce death, and radiotherapy using radiation to cause DNA damage and cancer cell death. (Figure 2C) Chemotherapy is also part of innovative therapies, which include targeted therapy using BRAF and MEK inhibitors to induce apoptosis and inhibit cancer cell proliferation, TLR-9 agonists to activate TCD8+ cells, adoptive cellular therapy involving the reinfusion of tumor-infiltrating lymphocytes (TILs), gene therapy using suicide genes to induce apoptosis, protease inhibitors, and immunotherapy with immune checkpoint inhibitors and cytokines to enhance T-cell activation against cancer cells.

## 6. Biotechnological Use of Serine Protease Inhibitors in the Anticancer Activity of Melanoma

Melanoma cells can alter the molecular integrity of cellular junctions, facilitating the metastasis process. Enzymes called proteases mediate this characteristic by catalyzing the degradation of proteins. In the carcinogenic process, proteases emerge as viable therapeutic targets for cancer treatment [162].

Elevated levels of various proteases underscore their importance in cancer progression. For example, increased levels of cysteine protease cathepsin B are associated with greater tumor invasion. A worse prognosis correlates with elevated metalloproteases, while reduced life expectancy links to increased levels of serine proteases, such as kallikreins [163,164,165]. The upregulation or activation of proteases creates an environment rich in proteolytic activity around the tumor. This phenomenon is critical. Tumor cells increase protease production to influence the local blood supply. They also facilitate cell migration across vessels and support movement within the cellular matrix during metastasis [166].

Nature widely distributes protease inhibitors, and plants serve as particularly abundant sources. Plants produce these inhibitors as a defense mechanism against injuries and attacks by insects and pathogens. These inhibitors also act as anti-metabolic proteins (anti-nutritional factors) and inhibit different stages of carcinogenesis [167]. The occurrence of protease inhibitors in legumes and cereal seeds has prompted plenty of investigations. Epidemiological evidence linking legumes with potential protective effects against certain types of cancer in vegetarian populations has further stimulated this interest. Researchers have explored using protease inhibitors to block tumor proliferation in both experimental models and cultured cells [166,168,169,170]. Therefore, protease inhibitors with cytotoxic potential as therapeutic agents have been used in chemotherapy protocols, yielding promising results for different melanoma lineages [171,172].

PIs are molecules that play a crucial role in regulating the activity of proteases, enzymes that catalyze the breakdown of peptide bonds in proteins. Excessive or dysregulated protease activity is associated with several diseases, including cancer, viral infections, inflammation, and neurodegenerative diseases [173]. Therefore, protease inhibitors have been widely studied for their potential therapeutic and biological effects, playing a critical role in therapies against various pathological conditions. Each class of protease inhibitors presents unique properties that can be harnessed to treat specific diseases, with ongoing research aimed at improving their efficacy and safety [173].

Serine protease inhibitors play a significant role in melanoma progression by modulating signaling pathways and tissue remodeling [174]. They also regulate the cell cycle, limit tumor growth, promote apoptosis, and reduce melanoma’s invasive and metastatic capacities. They impede the formation of new blood vessels that supply the tumor, which decreases the availability of essential nutrients for tumor growth [175]. Preclinical in vitro and in vivo studies demonstrate significant efficacy in reducing the vision of melanoma cells. However, clinical research is still in the early stages [176]. In vivo assays are a critical step in investigating the therapeutic potential of new compounds, allowing for the evaluation of efficacy, safety, and mechanism of action in a more complex biological context that more accurately mimics human physiology compared with in vitro assays. Table 2 shows the results of in vivo tests used to evaluate the anti-melanoma potential of some protease inhibitors.

Animal models are crucial for studying melanoma formation and progression, as well as for developing new therapeutic strategies for this aggressive cancer. They simulate the human organism’s complex environment, offering insights into tumor biology and the tumor microenvironment. These models help evaluate the efficacy and safety of new treatments [185]. Animal models facilitate the study of therapeutic transfers, helping to understand their synergies and toxicities. They enable the rapid evaluation of new drugs, optimizing time and resources in the development of therapeutic strategies [186]. The main models used to understand the dynamics and treatment of melanoma include Murine models—mice and rats are foundational in melanoma research because of their genetic and biological similarities to humans. These models are valuable because they replicate key aspects of human melanoma, including genetic mutations and the tumor microenvironment. Their genetic compatibility with humans makes them an ideal system for studying disease mechanisms and testing treatments [187]. In addition, Zebrafish (*Danio rerio*) offer a unique perspective on melanoma because of their rapid development, transparent embryos, and ease of genetic manipulation. Zebrafish are an emerging model for melanoma studies because of their rapid reproduction rate, embryo transparency, and ease of genetic manipulation [188].

Animal models have played an irreplaceable role in advancing our knowledge of melanoma and developing new therapeutic strategies. From understanding the disease’s biological mechanisms to evaluating new therapies, these models provide a critical platform for translational research. Despite their limitations, they remain essential tools for bridging the gap between laboratory research and clinical applications, improving treatment options and outcomes for patients with melanoma.

One notable example of an in vitro test is the Kunitz-type trypsin inhibitor, KTI-A, sourced from soybeans. This 24.2 kDa protein has demonstrated significant anti-proliferative effects in the melanoma cell line B16F10, highlighting soybeans as a potential source of bioactive compounds for pharmaceutical and industrial applications [189].

The protease inhibitor EcTI has shown cytotoxicity against various cancer cells, including melanoma, without affecting normal fibroblasts or stem cells. It impacts melanoma cell morphology, adhesion, migration, and invasion, and influences the activation of critical cancer-associated enzymes, matrix metalloproteinases (MMPs) [190,191,192]. CrataBL, a promising protein in melanoma treatment, displays substantial cytotoxicity against SK-MEL-28 cells. It effectively inhibits melanoma cell migration, achieving nearly 100% inhibition at 100 µM, and this effect is not solely due to cell death, suggesting additional mechanisms [193]. When combined with vemurafenib chemotherapy, CrataBL’s migratory inhibition is further enhanced, hinting at a potential synergy. Mechanistically, CrataBL disrupts vital cell signaling pathways, reducing proteins associated with cell adhesion, migration, and survival.

Additionally, CrataBL exhibits anti-inflammatory properties, reducing nitric oxide and inflammatory cytokines pivotal in tumor growth and metastasis. It induces late apoptosis in melanoma cells without significantly impacting necrosis, indicating an independent mechanism. CrataBL’s lectin properties enable it to bind to glycosaminoglycans [193], potentially interfering with crucial protease activities in melanoma. Overall, CrataBL’s inhibitory effects on melanoma encompass interference with matrix components, protease activities, and signaling proteins.

## 7. Future Perspectives

This study highlights the potential of protease inhibitors as valuable drugs for the treatment of melanoma. In terms of future prospects, the following points should be considered: The development of protease inhibitors offers a promising avenue for targeted therapy of melanoma. As our understanding of the molecular pathways of melanoma progression increases, researchers can design protease inhibitors with higher specificity and efficacy. Furthermore, the future of melanoma treatment may lie in combination therapies containing protease inhibitors. By combining protease inhibitors with other targeted drugs or immunotherapies, researchers hope to achieve synergistic effects and overcome potential resistance mechanisms.

Addressing drug resistance is a critical challenge in melanoma treatment. Future studies may focus on elucidating the mechanisms of resistance to protease inhibitors and developing strategies to circumvent or attenuate this resistance. Advances in genomics and proteomics are paving the way for personalized medicine approaches. Tailoring protease inhibitor therapy to individual patients based on their specific genetic and molecular characteristics may lead to more effective and personalized treatments.

Ongoing research may uncover new proteases and protease families that play critical roles in melanoma progression. Identification and characterization of these targets will expand the class of potential protease inhibitors. Future research could focus on developing innovative drug delivery systems to improve the targeted bioavailability and delivery of protease inhibitors to melanoma cells. This can improve therapeutic efficacy while minimizing off-target side effects.

Some protease inhibitors may have immunomodulatory properties that affect the tumor microenvironment and the immune response against melanoma. Understanding these effects may open new opportunities for combining therapies with immunotherapies. Taken together, the use of protease inhibitors in melanoma holds great promise for the future of targeted therapy in this challenging disease. Continued research, innovative drug development, and comprehensive clinical trials are critical to realizing the full potential of protease inhibitors as valuable additions to melanoma therapeutics.

## 8. Conclusions

Melanoma research is a dynamic and ongoing field, evident in the multitude of studies published over the years. These investigations have advanced our understanding of melanoma’s etiology, molecular biology, and specific subtypes, leading to improved early diagnosis and insights into disease progression risk factors. Efforts have also focused on refining melanoma staging for better risk stratification and treatment outcomes. The development of targeted therapies and immunotherapies represents significant progress, offering more efficient and personalized treatment options. Identifying risk factors and characterizing subtypes are crucial for early detection. Staging helps track disease progression, emphasizing the need for tailored treatment strategies. While current treatments have improved management, challenges like early resistance persist. To enhance the approach to metastatic melanoma, ongoing advancements in molecular understanding, exploration of new therapies, and strategies to overcome resistance are vital. The goal is to enhance patient survival and quality of life.

## Figures and Tables

**Figure 1 molecules-29-03891-f001:**
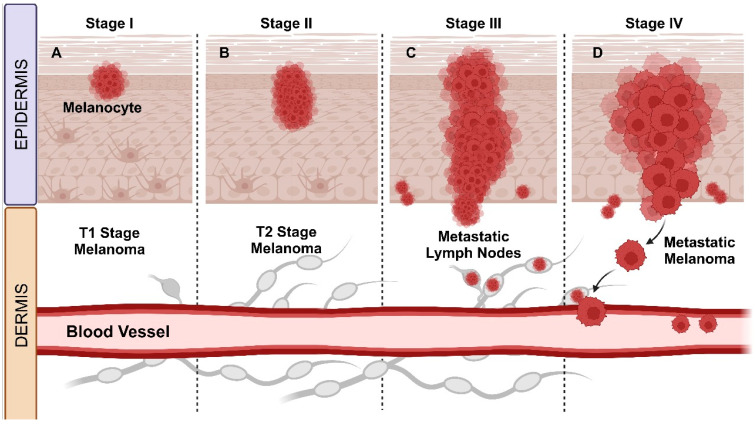
Melanoma staging phases. Primary melanomas are classified into four substages, T1–T4, based on tumor thickness and substage in the presence of ulceration (stages I and II). Patients with lymph node metastases (stage III), the clinical status of the lymph nodes (palpable or not), and the number of metastatic lymph nodes are the main prognostic factors. Stage IV melanoma is characterized by the spread of cancer beyond the skin and regional lymph nodes, reaching distant organs and tissues, through distant metastases. At this advanced stage, melanoma can spread to the liver, lungs, brain, bones, or other organs, which significantly increases the complexity of treatment and associated complications. Stage IV is generally considered to be an advanced disease stage, with a significant impact on prognosis and therapeutic options. (**A**) In first stage, the tumor is thin and less than 1 mm thick. (**B**) In stage II, the tumor is between 1 and 2 mm thick. (**C**) Presence of lymph node metastases. (**D**) Distant metastases are present. This figure was created with BioRender.

**Figure 2 molecules-29-03891-f002:**
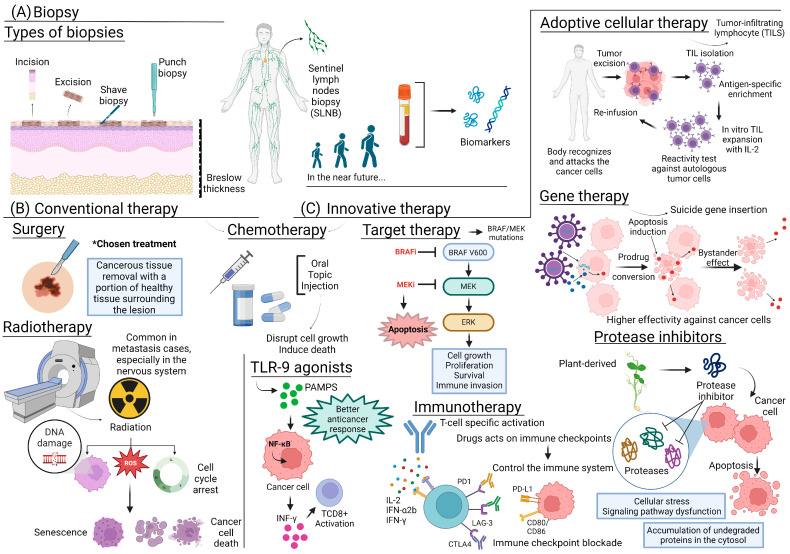
Overview of diagnostic and therapeutic approaches for cancer treatment.

**Table 1 molecules-29-03891-t001:** Melanoma classification based on the TNM system.

TNM Classification ^1^	Description
Primary staging of melanoma	
Tis	Melanoma in situ—The melanoma is confined to the outermost layer of the skin (epidermis) and has not invaded the basal layer. Also called carcinoma in situ.
T1	The tumor is thin, usually less than 1 mm thick. May or may not have ulcerated.
T2	The tumor is of moderate thickness, 1 to 2 mm in depth. It may or may not have ulcerated.
T3	The tumor is thicker, 2 to 4 mm deep. It may or may not have ulcerated.
T4	The tumor is very thick, over 4 mm deep. It may or may not have ulcerated.
Melanoma staging for regional lymph nodes	
N0	There is no evidence of regional lymph node metastasis.
N1	The melanoma has spread to a nearby lymph node but is micrometastasized (can only be detected with a microscope, cannot be seen or felt).
N2	The melanoma has spread to nearby lymph nodes and is considered macrometastatic metastasis (can be seen or felt).
N3	The melanoma has spread to nearby or distant lymph nodes, being multiple, grouped or with extranodal growth.
Metastatic Melanoma	
M0	There is no evidence of distant metastasis.
M1	The melanoma has spread to other parts of the body beyond the primary tumor area and regional lymph nodes.

^1^ Tables Adapted from Keung and Gershenwald [73].

**Table 2 molecules-29-03891-t002:** Main protease inhibitors used in melanoma and their biological characteristics.

Protease Inhibitor: Bowman-Birk Inhibitors (BBI)	
Source	Soybean (*Glycine max*) and other legumes.
Type of protease inhibited	Serine proteases (trypsin and chymotrypsin).
Biological effects	Inhibits cell proliferation and induce apoptosis in tumor cells. Demonstrates anticancer, anti-inflammatory, and antioxidant properties. Protects against oxidative damage and inflammation and may help prevent cancers such as melanoma.
Research status	Studied in preclinical and early clinical models; showed efficacy in reducing tumor growth and cell protection.
References	Lyu et al. [177]; Chen et al. [178]; Gitlin-Domagalska et al. [179]; Sato et al. [180].
Protease Inhibitor: Soybean Trypsin Inhibitor (Kunitz-Type)	
Source	Soybean (*Glycine max*) and other legumes.
Type of protease inhibited	Serine proteases (trypsin and chymotrypsin).
Biological effects	Reduces tumoral cell proliferation and blocks cell invasion, especially in melanoma cells. Affects angiogenesis, decreasing the formation of blood vessels that feed tumors.
Research status	In vitro and in vivo studies demonstrate efficacy in reducing melanoma cell growth.
References	Shigetomi et al. [181]; Maria et al. [182]; Ranasinghe et al. [183].
Protease Inhibitor: Potato Trypsin Inhibitor	
Source	Potato (*Solanum tuberosum*).
Type of protease inhibited	Serine protease (trypsin).
Biological effects	Affects cell cycle regulation, inhibiting cancer cell growth. Reduces tumoral metastasis and invasion in melanoma models by stabilizing the extracellular matrix. Demonstrates anti-inflammatory properties, which may be useful in autoimmune diseases.
Research status	Promising results in laboratory studies and animal model. Focused on formulation for clinical use.
References	Liu et al. [184]

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
