# Peer review of "Cutaneous Melanoma: An Overview of Physiological and Therapeutic Aspects and Biotechnological Use of Serine Protease Inhibitors"

_molecules, 2024, doi:10.3390/molecules29163891_

Round 1

Reviewer 1 Report

Comments and Suggestions for Authors

The manuscript of De Araùjo AP et al., offers a very detailed review about melanoma, well describing each sub-group from an epidemiological and histopathological point of view. They also accurately described therapeutics approaches and tumor incidences worldwide. Although the article is excellent, the authors should address some comments about the molecular mechanism leading to melanoma, al well as the cell of origin (neural crest). This should improve the quality of their work.

Major Comments

1) Describe in better detail the cell of origin of melanoma.

2) For each melanoma subtype, please describe better (with few words/lines) the molecular mechanism.

3) Several animal models are used to better understand melanoma formation (e.g. mouse, zebrafish). Please introduce and describe (in few lines) these models and their impact on therapeutic strategies.

Minor Comments

33/34: in the Introduction, authors write: “It affects melanocytes, derived derivative cells of the neural crest…”. Since the article is centered on melanoma, the cells of origin of melanoma are melanocyte, a neural crest derivative that are located in the basal layer of epidermis. This must be described better.

149/150: Authors write: “Mutations in LMM vary from mucosal melanoma (SSM), including genetic changes in NF1, BRAFV600K, non-V600E mutations, NRAS, and to a lesser extent, KIT”. Please describe better the impact of these mutations in the cell of origin (Neural Crest). As an example, NF1 is also involved in Neurofibromatosis (NF) – Malignant Peripheral Nerve Shear Tumors (MPNSTs).

Author Response

Editor Office

Molecules

Manuscript ID: molecules-3116056

Please find attached a revised version of our manuscript “Cutaneous melanoma: An overview of physiological and therapeutic aspects and biotechnological use of serine protease inhibitors”.

The comments of the reviewers were enlightening, improving the quality of our paper. In the next pages, are our present the changes suggested by the reviewers.

#REVIEWER 1:

Comments and Suggestions for Authors

The manuscript of De Araùjo AP et al., offers a very detailed review about melanoma, well describing each sub-group from an epidemiological and histopathological point of view. They also accurately described therapeutics approaches and tumor incidences worldwide. Although the article is excellent, the authors should address some comments about the molecular mechanism leading to melanoma, as well as the cell of origin (neural crest). This should improve the quality of their work.

Major Comments

  • Describe in better detail the cell of origin of melanoma.

Response: Considering the suggestion and revising the first paragraph, it has been completely rewritten.

Now reads, page 2, line 49:

“Melanoma develops when melanocytes, which produce pigment, grow abnormally in the basal layer of the epidermis. Melanocytes have melanosomes with two types of pigments: pheomelanin (yellow/red) and eumelanin (brown/black). Eumelanin is the primary photoprotective pigment in the skin, while pheomelanin generates reactive oxygen species under UV radiation, leading to DNA damage [3].

Melanogenesis can be stimulated in a paracrine, autocrine, or intracrine manner. The melanocyte-stimulating binding hormone (α-MSH) promotes the production of eumelanin by binding to the Melanocortin 1 Receptor (CM1R), which then triggers an increase in Intracellular Cyclic Adenosine Monophosphate (cAMP) levels and activates the cAMP-response-element-binding protein (CREB). Along with SOX10 (Sex-Determining Region Y-box 10), and PAX3 (Paired Box 3) transcription factors this, leads to activation of MITF (Microphthalmia-associated Transcription Factor), resulting in increased expression of crucial enzymes for melanogenesis from amino acid tryptophan [3].

Conversely, melanocytes are derived from the Neural Crest (NC), a group of versatile precursor cells between the neural tube and the ectoderm. This region is also responsible for generating neurons, glial cells, and various other cell types. During early embryogenesis, neuroectodermal cells form the neural plate, which folds and closes to form the neural tube. NC cells migrate, in a delamination process, along the dorsolateral route to the epidermis, where they differentiate into melanocytes in the basal layer. A melanocytes subpopulation may originate from cells that migrated in the ventral route, proliferating in the dermis before attaching to the epidermis [4].”

  • For each melanoma subtype, please describe better (with few words/lines) the molecular mechanism.

Response: The authors are in according with reviewer’s suggestions, seeking to improve the exposure of the data obtained and the interpretation of this review, changes were realized throughout of the maintext.

Now reads, page 5, line 198:

“The mitogen-activated protein kinase (MAPK) pathway, responsible for transmitting extracellular signals to the nucleus and regulating cell proliferation and differentiation, is frequently dysregulated in cutaneous melanoma [39]. Approximately 90% of cutaneous melanomas exhibit abnormal activation of the mitogen-activated protein kinase (MAPK) pathway, primarily due to mutations in the BRAF gene. Somatic mutations in BRAF are prevalent in 37% to 60% of melanomas, most commonly occurring at position 600. About 80% of these mutations are of the V600E type, involving a substitution of valine with glutamic acid, while 5% to 12% are V600K mutations, where lysine substitutes valine. The protein kinase encoded by the BRAF gene regulates the phosphorylation of mitogen-activated protein kinase kinase (MEK) and maintains the protein in an inactive state through hydrophobic [40,41].”

In the topic on Desmoplastic melanoma, two paragraphs were added on the molecular mechanism of this type of melanoma, page 5, line 225:

“Desmoplastic melanoma has a distinct genetic profile compared to other melanoma subtypes. Mutations in the B-Raf proto-oncogene (BRAF) and neuroblastoma RAS viral oncogene homolog (NRAS), common in other melanomas, are less frequent in this variant. However, mutations in the neurofibromin 1 (NF1) gene are more prevalent, leading to the activation of the RAS-mitogen-activated protein kinase (MAPK) pathway, which promotes cell proliferation. Additionally, alterations in the tumor protein p53 (TP53) gene, resulting in the loss of p53 tumor suppressor function, are commonly observed, contributing to genomic instability and tumor progression [49, 50].

By inactivating NF1, the activation of the mitogen-activated protein (MAP) kinase pathway is prolonged, enhancing cellular proliferation [51]. Common mutations in BRAF, and neuroblastoma RAS viral oncogene homolog (NRAS), typically present in other melanomas, are absent in this context. However, genetic alterations have been identified in NF1, casitas B-lineage lymphoma (CBL), erb-b2 receptor tyrosine kinase 2 (ERBB2), MAP kinase kinase 1 (MAP2K1), MAP kinase kinase kinase 1 (MAP3K1), BRAF, epidermal growth factor receptor (EGFR), protein tyrosine phosphatase non-receptor type 11 (PTPN11), MET proto-oncogene (MET), Ras-related C3 botulinum toxin substrate 1 (RAC1), SOS Ras/Rac guanine nucleotide exchange factor 2 (SOS2), NRAS, and phosphatidylinositol-4,5-bisphosphate 3-kinase catalytic subunit alpha (PIK3CA). Some of these are potential candidates for targeted therapies [43]. Treatment options include surgical excision, sentinel lymph node biopsy, adjuvant radiotherapy to the tumor bed, and systemic chemotherapy. Recently, DM has been observed to be associated with multiple genetic mutations and has shown responsiveness to targeted therapies [47, 48].”

In the topic acral melanoma, modifications were added, page 6, line 256

“Unlike other forms of cutaneous melanoma, acral melanoma is not influenced by risk factors like UV rays exposure, fair skin type, family or personal history of melanoma, or pre-existing melanocytic nevi [53, 54, 56]. This type of melanoma rarely presents mutations in the BRAF and NRAS genes, common in other melanomas, but has a high frequency of mutations in the KIT gene, which can activate the MAPK signaling pathway. n addition, acral melanoma frequently exhibits complex structural alterations, including amplifications and deletions of chromosomes, particularly affecting the cyclin D1 (CCND1), cyclin-dependent kinase 4 (CDK4), and telomerase reverse transcriptase (TERT) genes [56].”

In the Spitz Melanoma topic, a paragraph was also added about the mechanism of action of this melanoma, page 6, line 289.

“Genetic fusions involving anaplastic lymphoma kinase (ALK), neurotrophic receptor tyrosine kinase 1 (NTRK1), ROS proto-oncogene 1 (ROS1), rearranged during transfection (RET), and B-Raf proto-oncogene serine/threonine kinase (BRAF) genes are frequently observed in this type of melanoma. These fusions lead to the continuous activation of signaling pathways, such as the mitogen-activated protein kinase (MAPK) pathway, which support cell growth and survival. Mutations in the Harvey rat sarcoma viral oncogene homolog (HRAS) gene are also observed, contributing to the activation of the phosphatidylinositol 3-kinase (PI3K)-AKT pathway. These fusions and mutations lead to unique biological behavior, with the potential for rapid growth but a lower metastasis rate when compared to other melanomas. Understanding these molecular features is crucial for developing targeted and personalized treatments for patients with Spitz melanoma [58, 59, 60].”

And in the Mucosal melanoma topic, a paragraph was also added about the molecular mechanism of this type of melanoma, page 8, line 339:

Recent genetic sequencing studies have identified alterations in neuroblastoma RAS viral oncogene homolog (NRAS), BRAF, neurofibromin 1 (NF1), KIT, splicing factor 3b subunit 1 (SF3B1), tumor protein p53 (TP53), and sprouty-related EVH1 domain-containing protein 1 (SPRED1), revealing potential targeted therapeutic strategies for MM. Patients suffering from this condition display a reliance on the MAPK pathway, although their mutation patterns vary from those seen in cutaneous melanoma (CM). NRAS and BRAF mutations are less common in MM, while SF3B1 mutations and KIT alterations are more frequent. Given the lower prevalence of targetable BRAF mutations in MM, it is necessary to validate targets in other alterations within the MAPK pathway [66].”

And melanoma arising in a congenital nevus topic, a paragraph was also added about the molecular mechanism of this type of melanoma, page 8, line 364:

“Melanoma arising from a congenital nevus is characterized by activating mutations in the neuroblastoma RAS viral oncogene homolog (NRAS) gene, often involving the loss of the normal allele and amplification of the mutant NRAS. These changes promote the MAPK pathway activation, contributing to malignant transformation. Additional mutations, such as changes in TP53 and CDKN2A, are required for tumor progression. Despite the infrequency of BRAF mutations in this melanoma type, genotyping for both NRAS and BRAF is recommended as a result of the effectiveness of available inhibitors. Large congenital nevi often present with genomic instability, which highlights the importance of monitoring and early intervention [69, 70]. In melanomas linked to giant congenital nevi, telomerase reverse transcriptase (TERT) expression can be upregulated through epigenetic modifications, specifically methylation, while the tumors maintain the wild-type genotype [71, 72].”

  • Several animal models are used to better understand melanoma formation (e.g. mouse, zebrafish). Please introduce and describe (in few lines) these models and their impact on therapeutic strategies.

Response: The authors are in agreement with reviewer, and two paragraph on use of animal model and zebrafish was included in topic 6.

Now reads, page 22, line 874:

“Animal models are crucial for studying melanoma formation and progression, as well as for developing new therapeutic strategies for this aggressive cancer. They simulate the human organism’s complex environment, offering insights into tumor biology and the tumor microenvironment. These models help evaluate the efficacy and safety of new treatments [185]. Animal models facilitate the study of therapeutic transfers, helping to understand their synergies and toxicities. They enable the rapid evaluation of new drugs, optimizing time and resources in the development of therapeutic strategies [186]. The main models used to understand the dynamics and treatment of melanoma include: Murine models - Mice and rats are foundational in melanoma research because of their genetic and biological similarities to humans. These models are valuable because they replicate key aspects of human melanoma, including the genetic mutations and tumor microenvironment. Their genetic compatibility with humans makes them an ideal system for studying disease mechanisms and testing treatments [187]. In addition, Zebrafish (Danio rerio) offer a unique perspective on melanoma because of their rapid development, transparent embryos, and ease of genetic manipulation. Zebrafish are an emerging model for melanoma studies, as a result of their rapid reproduction rate, embryo transparency, and ease of genetic manipulation [188].

Animal models have played an irreplaceable role in advancing our knowledge of melanoma and developing new therapeutic strategies. From understanding the disease’s biological mechanisms to evaluating new therapies, these models provide a critical platform for translational research. Despite their limitations, they remain essential tools for bridging the gap between laboratory research and clinical applications, improving treatment options and outcomes for patients with melanoma.

Minor Comments

33/34: in the Introduction, authors write: “It affects melanocytes, derived derivative cells of the neural crest…”. Since the article is centered on melanoma, the cells of origin of melanoma are melanocyte, a neural crest derivative that are located in the basal layer of epidermis. This must be described better.

 Response: The authors are in according with reviewer and in order to improve the information, the manuscript was revised.

Now reads, page 2, line 63:

“Conversely, melanocytes are derived from the Neural Crest (NC), a group of versatile precursor cells between the neural tube and the ectoderm. This region is also responsible for generating neurons, glial cells, and various other cell types. During early embryogenesis, neuroectodermal cells form the neural plate, which folds and closes to form the neural tube. NC cells migrate, in a delamination process, along the dorsolateral route to the epidermis, where they differentiate into melanocytes in the basal layer. A melanocytes subpopulation may originate from cells that migrated in the ventral route, proliferating in the dermis before attaching to the epidermis [4].”

149/150: Authors write: “Mutations in LMM vary from mucosal melanoma (SSM), including genetic changes in NF1, BRAFV600K, non-V600E mutations, NRAS, and to a lesser extent, KIT”. Please describe better the impact of these mutations in the cell of origin (Neural Crest). As an example, NF1 is also involved in Neurofibromatosis (NF) – Malignant Peripheral Nerve Shear Tumors (MPNSTs).

Response: The authors are in according with reviewer and to improve the information.

Now reads, page 4, line 184:

“Genetic changes in neurofibromin 1 (NF1), B-Raf proto-oncogene (BRAF)V600K, non-V600E (glutamic acid substitution) mutations, neuroblastoma RAS viral oncogene homolog (NRAS), and, to a lesser extent, KIT proto-oncogene receptor tyrosine kinase (KIT), are among the varying mutations in lentigo maligna melanoma (LMM) compared to superficial spreading melanoma (SSM) [39]. The NF1 gene makes individuals more susceptible to multiple tumors, particularly those that arise from the neural crest. This autosomal dominant disorder has a high mutation rate, with approximately 50% being new mutations. Neurofibromin, the protein encoded by the NF1 gene, plays a crucial role in negatively regulating the Ras/MAPK pathway, which enhances cell proliferation and suppresses apoptosis. A deficiency in neurofibromin is linked to the development of various tumors, both benign and malignant. Individuals with NF1 are at a higher risk of developing optic pathway glioma, glioblastoma, Malignant Peripheral Nerve Sheath Tumor (MPNST), breast cancer, sarcoma, leukemia, Gastrointestinal Stromal Tumor (GIST), pheochromocytoma, duodenal carcinoid tumor, and melanoma [40].”

Thank you for your consideration of this manuscript.

Sincerely,

Profa. Dra. Maria Lígia Rodrigues Macedo

Laboratório de Purificação de Proteínas e Suas Funções Biológicas (LPPFB)

Universidade Federal de Mato Grosso do Sul

Postal Box 549, Campo Grande, Mato Grosso do Sul State, 79070-900, Brazil.

Reviewer 2 Report

Comments and Suggestions for Authors

This work is of interest but certain modifications should be made:

1. The authors should review the mutation profile of this group of patients compared to other groups, as well as the expression profilin date.

2. More comprehensive experimental therapeutic data should be mentioned.

Comments on the Quality of English Language

Acceptable.

Author Response

Editor Office

Molecules

Manuscript ID: molecules-3116056

Dear Reviewer 2

Please find attached a revised version of our manuscript “Cutaneous melanoma: An overview of physiological and therapeutic aspects and biotechnological use of serine protease inhibitors”.

The comments of the reviewers were enlightening, improving the quality of our paper. In the next pages, are our present the changes suggested by the reviewers.

Comments and Suggestions for Authors

This work is of interest but certain modifications should be made:

  1. The authors should review the mutation profile of this group of patients compared to other groups, as well as the expression profilin date.

Response: The authors are in according with reviewer’s suggestions, seeking to improve the exposure of the data obtained and the interpretation of this review, changes were realized throughout of the maintext. Where each mutation profile was described for each melanoma subtype.

Now reads, page 5, line 198:

“The mitogen-activated protein kinase (MAPK) pathway, responsible for transmitting extracellular signals to the nucleus and regulating cell proliferation and differentiation, is frequently dysregulated in cutaneous melanoma [39]. Approximately 90% of cutaneous melanomas exhibit abnormal activation of the mitogen-activated protein kinase (MAPK) pathway, primarily due to mutations in the BRAF gene. Somatic mutations in BRAF are prevalent in 37% to 60% of melanomas, most commonly occurring at position 600. About 80% of these mutations are of the V600E type, involving a substitution of valine with glutamic acid, while 5% to 12% are V600K mutations, where lysine substitutes valine. The protein kinase encoded by the BRAF gene regulates the phosphorylation of mitogen-activated protein kinase kinase (MEK) and maintains the protein in an inactive state through hydrophobic [40,41].”

In the topic on Desmoplastic melanoma, two paragraphs were added on the molecular mechanism of this type of melanoma, page 5, line 225:

“Desmoplastic melanoma has a distinct genetic profile compared to other melanoma subtypes. Mutations in the B-Raf proto-oncogene (BRAF) and neuroblastoma RAS viral oncogene homolog (NRAS), common in other melanomas, are less frequent in this variant. However, mutations in the neurofibromin 1 (NF1) gene are more prevalent, leading to the activation of the RAS-mitogen-activated protein kinase (MAPK) pathway, which promotes cell proliferation. Additionally, alterations in the tumor protein p53 (TP53) gene, resulting in the loss of p53 tumor suppressor function, are commonly observed, contributing to genomic instability and tumor progression [49, 50].

By inactivating NF1, the activation of the mitogen-activated protein (MAP) kinase pathway is prolonged, enhancing cellular proliferation [51]. Common mutations in BRAF, and neuroblastoma RAS viral oncogene homolog (NRAS), typically present in other melanomas, are absent in this context. However, genetic alterations have been identified in NF1, casitas B-lineage lymphoma (CBL), erb-b2 receptor tyrosine kinase 2 (ERBB2), MAP kinase kinase 1 (MAP2K1), MAP kinase kinase kinase 1 (MAP3K1), BRAF, epidermal growth factor receptor (EGFR), protein tyrosine phosphatase non-receptor type 11 (PTPN11), MET proto-oncogene (MET), Ras-related C3 botulinum toxin substrate 1 (RAC1), SOS Ras/Rac guanine nucleotide exchange factor 2 (SOS2), NRAS, and phosphatidylinositol-4,5-bisphosphate 3-kinase catalytic subunit alpha (PIK3CA). Some of these are potential candidates for targeted therapies [43]. Treatment options include surgical excision, sentinel lymph node biopsy, adjuvant radiotherapy to the tumor bed, and systemic chemotherapy. Recently, DM has been observed to be associated with multiple genetic mutations and has shown responsiveness to targeted therapies [47, 48].”

In the topic acral melanoma, modifications were added, page 6, line 256

“Unlike other forms of cutaneous melanoma, acral melanoma is not influenced by risk factors like UV rays exposure, fair skin type, family or personal history of melanoma, or pre-existing melanocytic nevi [53, 54, 56]. This type of melanoma rarely presents mutations in the BRAF and NRAS genes, common in other melanomas, but has a high frequency of mutations in the KIT gene, which can activate the MAPK signaling pathway. n addition, acral melanoma frequently exhibits complex structural alterations, including amplifications and deletions of chromosomes, particularly affecting the cyclin D1 (CCND1), cyclin-dependent kinase 4 (CDK4), and telomerase reverse transcriptase (TERT) genes [56].”

In the Spitz Melanoma topic, a paragraph was also added about the mechanism of action of this melanoma, page 6 and 7, line 289.

“Genetic fusions involving anaplastic lymphoma kinase (ALK), neurotrophic receptor tyrosine kinase 1 (NTRK1), ROS proto-oncogene 1 (ROS1), rearranged during transfection (RET), and B-Raf proto-oncogene serine/threonine kinase (BRAF) genes are frequently observed in this type of melanoma. These fusions lead to the continuous activation of signaling pathways, such as the mitogen-activated protein kinase (MAPK) pathway, which support cell growth and survival. Mutations in the Harvey rat sarcoma viral oncogene homolog (HRAS) gene are also observed, contributing to the activation of the phosphatidylinositol 3-kinase (PI3K)-AKT pathway. These fusions and mutations lead to unique biological behavior, with the potential for rapid growth but a lower metastasis rate when compared to other melanomas. Understanding these molecular features is crucial for developing targeted and personalized treatments for patients with Spitz melanoma [58, 59, 60].”

And in the Mucosal melanoma topic, a paragraph was also added about the molecular mechanism of this type of melanoma, page 8, line 339:

Recent genetic sequencing studies have identified alterations in neuroblastoma RAS viral oncogene homolog (NRAS), BRAF, neurofibromin 1 (NF1), KIT, splicing factor 3b subunit 1 (SF3B1), tumor protein p53 (TP53), and sprouty-related EVH1 domain-containing protein 1 (SPRED1), revealing potential targeted therapeutic strategies for MM. Patients suffering from this condition display a reliance on the MAPK pathway, although their mutation patterns vary from those seen in cutaneous melanoma (CM). NRAS and BRAF mutations are less common in MM, while SF3B1 mutations and KIT alterations are more frequent. Given the lower prevalence of targetable BRAF mutations in MM, it is necessary to validate targets in other alterations within the MAPK pathway [66].”

And melanoma arising in a congenital nevus topic, a paragraph was also added about the molecular mechanism of this type of melanoma, page 8, line 364:

“Melanoma arising from a congenital nevus is characterized by activating mutations in the neuroblastoma RAS viral oncogene homolog (NRAS) gene, often involving the loss of the normal allele and amplification of the mutant NRAS. These changes promote the MAPK pathway activation, contributing to malignant transformation. Additional mutations, such as changes in TP53 and CDKN2A, are required for tumor progression. Despite the infrequency of BRAF mutations in this melanoma type, genotyping for both NRAS and BRAF is recommended as a result of the effectiveness of available inhibitors. Large congenital nevi often present with genomic instability, which highlights the importance of monitoring and early intervention [69, 70]. In melanomas linked to giant congenital nevi, telomerase reverse transcriptase (TERT) expression can be upregulated through epigenetic modifications, specifically methylation, while the tumors maintain the wild-type genotype [71, 72].”

  1. More comprehensive experimental therapeutic data should be mentioned.

Response: The authors are in according with reviewer and in order to improve the information, the manuscript was revised and new therapeutic data were included in topic 5.

Now reads, page 12, topic 5.1.1, line 497:

“Recently, several clinical reports demonstrating the efficacy of Mohs micrographic surgery in melanoma therapy have considered factors such as patient safety for those diagnosed with melanoma in situ (MIS) and invasive melanoma (IM), low treatment costs, tumor staging risks, and potential failure in sentinel lymph node biopsy (SLNB). These studies suggest that Mohs surgery yields result like or even better than local tumor excision [95]. On the other hand, other studies, such as those by Huerta et al. [96], suggest that current evidence does not support the equivalence of Mohs micrographic surgery to wide excision for cutaneous melanoma. A more detailed, individualized evaluation is always crucial. Additionally, given the complex context of melanoma, recent studies have encouraged combined and systemic approaches to increase the likelihood of successful therapy [97].”

Now reads, page 13, topic 5.1.2, line 517:

“In the therapeutic context, radiotherapy can be used for distant metastases and subcutaneous recurrences to aid in disease control and symptom relief. This method can be associated with conditions such as hyperthermia and even immunotherapy. Boron neutron capture therapy might be an option alongside standard treatments, although its clinical application is still limited [99]. Radiotherapy has demonstrated variable efficacy in treating local melanoma, particularly in cases of brain metastases. Consequently, there is an increasing encouragement for combined therapies [100]. Recent clinical data and trials have shown improvements with combinations such as intensity-modulated therapy combined with stereotactic surgery and immunotherapy, especially with checkpoint inhibitors (discussed in subsequent sections), which have exhibited synergistic potential [101, 102, 103]. Despite these promising results, radioinduced resistance and associated adverse effects remain significant challenges that need further investigation and resolution.”

Now reads, page 14, topic 5.2.1, line 556:

“The topical dermal route is easy to administer, non-invasive, and minimizes systemic effects, thus enhancing patient compliance. Semisolid dosage forms, such as gels, are particularly advantageous due to their ability to adhere well to the application site, prolong the release of active ingredients, and provide excellent spreadability. Gels also tend to have fewer long-term stability issues compared to other semisolid forms and are relatively straightforward to formulate, accommodating a variety of active compounds [106]. According to Cancer Research UK, topical treatments are applied directly to the lesion site and are generally administered once or twice a day for 3 to 4 weeks, following medical advice [107].”

Now reads, page 14, topic 5.2.2, line 587:

“Targeted therapy is a cancer treatment that specifically targets proteins and genes responsible for regulating the growth, division, and spread of cancer cells. As our understanding of genetic mutations and protein mechanisms in cancer cells advances, researchers can develop more precise treatments [107]. In melanoma treatment, targeted therapy is particularly revolutionary because it causes less damage to healthy cells. These treatments focus on specific aspects of the immune system or on problematic genes and molecules, such as inhibiting the mitogen-activated protein kinase (MAPK) pathway in tumors with BRAF or MEK mutations [116], or targeting pathways like cyclin-dependent kinases, such as CDK4/6 [117].”

And, page 15, line 612:

“BRAFi and MEKi are capsules or pills normally taken once or twice a day [123].

Phase 1 and 2 clinical studies with the BRAF inhibitor vemurafenib showed a 50% response rate in patients diagnosed with the BRAF V600E mutation. Additionally, a phase 3 randomized study comparing vemurafenib with dacarbazine in untreated metastatic melanoma demonstrated that vemurafenib resulted in greater overall survival and reduced disease progression [124]. In the Columbus study conducted by Dummer et al. [125], the combination of the BRAF inhibitor encorafenib and the MEK inhibitor binimetinib was compared with vemurafenib alone. This combination proved to be more effective, showing increased overall survival for patients with the BRAF mutation. Similarly, the phase 3 coBRIM study, published by Ascierto et al. [126], demonstrated the efficacy of combining cobimetinib with vemurafenib for treating patients with advanced melanoma with the V600 mutation. The efficacy of this pharmacological combination was further confirmed in phase 3 randomized studies, COMBI-d [127] and COMBI-v [128], which assessed the quality of life of patients receiving the combined therapy of dabrafenib and trametinib versus vemurafenib. These studies consistently showed that the combination therapy resulted in healthier patients and more effective treatment outcomes. The COMBI-r study, based on daily clinical practice, further confirmed these results [129].”

And, page 16, line 636:

“Here, we review the main mutations associated with cutaneous melanoma, focusing on the BRAF, MEK, and KIT genes. We provide a detailed comparison of mutation rates and types among patients with cutaneous melanoma, other melanoma subtypes such as acral and mucosal melanoma, and different ethnic groups. Recent studies have shown that persistent tumor mutation burden (pTMB) is directly associated with improved responses to immune checkpoint blockade therapy. Xu et al. [133] developed a melanoma model based on pTMB, which demonstrated a higher incidence of genetic alterations and greater immune cell infiltration in the high-pTMB group. Another study identified the COL4A3 gene as a significant therapeutic target in UV-related cutaneous melanoma, as it inhibits cell proliferation and migration [134]. Li et al. [135] suggested using immunological and transcriptomic profiling to develop diagnostic and prognostic models for cutaneous melanoma, identifying crucial genes involved in immune response and melanoma progression.

Mutations in the BRAF, NRAS, and KIT genes activate various signaling pathways, notably the MAPK/ERK pathway, which can influence cytoskeletal organization. This activation can lead to actin reorganization through modulators like profilin, impacting the migratory and invasive abilities of tumor cells [133]. Constitutive activation of these pathways, such as the BRAFV600E mutation, can elevate the expression and activity of cytoskeletal regulatory proteins, including profilin. Changes in profilin expression can thus modulate tumor cell invasion and metastasis, processes heavily dependent on actin dynamics [136]. Studies have shown that profilin regulation directly affects cellular invasion and proliferation, making profilin and its regulators potential therapeutic targets [134]. For instance, overexpression of COL4A3, a gene identified as significant in melanoma, can influence profilin regulation, actin dynamics, and the invasive capacity of tumor cells [135].”

Now reads, page 17, topic 5.2.3, line 702:

“In a phase 3 study examining the combination of atezolizumab, cobimetinib, and vemurafenib, it was demonstrated that this combination was more effective than the individual drugs, particularly for patients with advanced BRAFV600 mutations [143]. Another phase 2 study investigated the safety of combining a PD-L1 inhibitor (lambrolizumab) with an anti-CTLA-4 inhibitor (ipilimumab). This study showed greater tumor regression in patients who received the combination therapy, even in cases where tumors initially developed when treated only with ipilimumab [144]. Later, the same research group published an article analyzing patient response to a combination of nivolumab (a PD-1 inhibitor) and ipilimumab (a CTLA-4 inhibitor). This combination therapy resulted in significant improvements in overall survival and progression-free survival compared to individual therapies. Although the treatment led to more grade 3 and 4 adverse effects, these were manageable and allowed for the maintenance of patients' quality of life [145]. The data confirm that combined therapy with CTLA-4 and PD-1/PD-L1 inhibitors is effective, enhancing both patient survival and quality of life.”

Now reads, page 18, topic 5.2.4, line 735:

“Recent clinical studies have explored the anti-melanoma potential of treatments involving TLR-9 agonists. Vidutolimab, when combined with pembrolizumab, achieved a 25% response rate in patients with PD-1 refractory melanoma. Additionally, the combination demonstrated a safe profile [151]. Another trial investigated the compound CMP-001, also in combination with pembrolizumab, through a phase 1b study with two parts. This combination was administered intratumorally and showed good tolerance as well as a significant local tumor response, highlighting the therapy’s potential for combined administration [152].”

Now reads, page 19, topic 5.2.5, line 757:

“Recently, the FDA approved adoptive cell therapy using tumor-infiltrating lymphocytes (TILs), specifically lifecycle, for the treatment of advanced melanoma. The phase 3 trial, conducted for the first time in 2022, demonstrated a 50% reduction in solid tumors compared to standard therapy [156]. The FDA’s approval of this therapy, along with its growing acceptance among clinicians, represents a significant advancement in improving patient response rates and survival.”

Now reads, page 19, topic 5.2.6, line 779:

“Phase 1/2 studies investigated the efficacy of DF6002, an oncolytic agent, in combination with nivolumab for patients with advanced or metastatic melanoma. This study includes elements of suicide gene therapy, aiming to enhance the antitumor response while minimizing side effects ([160]. Another study by the Henry Ford Health Institute examined a genetically modified adenovirus incorporating suicide genes. These genes are designed to interfere with DNA replication, promoting cell death, particularly when used in conjunction with chemotherapy or radiotherapy [161].”

Thank you for your consideration of this manuscript.

Sincerely,

Profa. Dra. Maria Lígia Rodrigues Macedo

Laboratório de Purificação de Proteínas e Suas Funções Biológicas (LPPFB)

Universidade Federal de Mato Grosso do Sul

Postal Box 549, Campo Grande, Mato Grosso do Sul State, 79070-900, Brazil.

Reviewer 3 Report

Comments and Suggestions for Authors

Boleti et al. wrote a review about cutaneous melanoma. The authors talked about the epidemiology and subtypes of melanoma. Then the authors discussed the staging phases and therapeutic strategies including the treatment with protease inhibitors. This will be a useful review for cutaneous melanoma. However, the current manuscript lacks figure illustration and needs more work on organization and typo errors. The following issues should also be resolved to improve the presentation of this review before consideration for publication.

Comments:

1.     Lines 43-44: "ABCDE" signs were mentioned here. Only "A", "B", "C", and "D" were explained. Just curious, what does "E" stand for? Or "E" just make it rhyme.

2.     Line 94: “global incidence has increased, especially in age groups and men.” Did you mean a group of specific age here?

3.     Multiple typos: line 102: “doing”, which organ did you want to mention?; line 131: “DSC”, I guess this should be “CSD”; Line 229: “bsuch as”; line 585: “IPs”: This should be protease inhibitors.

4.     Line 256-258: “low-CSD (Copy-Number-Signature-low)”: In lines 130-131, CSD was defined as cumulated solar damage. However, line 256 shows that CSD means Copy-Number-Signature-low. Do you mean low-CNS here? Line 258: Were “disguised” meant to be used here?

5.     Paragraph spacing is not consistent across the manuscript. e.g. lines 271-277 and lines 355-363.

6.     The numbering of sections 3, 4, and 5 is not consistent and messed up. Each section should have its separate numbering and need to be consistent in the same section.

7.     Section 5.2.1 and 5.2.2: What are the definitions of topical treatment and targeted therapy and how to perform them? Without a clear explanation, it seems that they are like regular chemotherapy.

8.     Protease inhibitors can be small molecules, small proteins, or polypeptides. What are the protease inhibitors that have been studied so far? The authors mentioned a few small protein inhibitors in the paper. Are there any clinical trials using protease inhibitors? If the authors want to emphasize protease inhibitors here, more elaboration should be done. For example, a table summarizing the protease inhibitors with their types, effects, etc will be helpful.

9.     Most of this manuscript is about Melanoma subtypes, staging strategies, and treatments. Protease inhibitors were only discussed in section 6. However, the title, abstract, and future perspectives seem to emphasize the application of protease inhibitors in treating melanoma. Is there a particular reason to emphasize the protease inhibitors? The current manuscript does not show enough necessity or importance of protease inhibitors. For example, the use of protease inhibitors looks a lot like chemotherapy except that the protease inhibitors applied seem to be small proteins.

10.  It will be helpful if the authors can add more figure illustrations to ease the understanding of the main points of the manuscript. The current manuscript only has one figure. For a review paper, this is not enough to help readers understand the manuscript. E.g. figures showing different melanoma subtypes, figures illustrating topical treatments and targeted therapy, and figures showing protease inhibitor treatment and action of mechanism.

Comments on the Quality of English Language

A few typo errors were found in the manuscript. I mentioned a few in the comments. Please proofread the manuscript to solve the typos and grammar errors as needed.

Author Response

Editor Office

Molecules

Manuscript ID: molecules-3116056

Dear Reviewer 3

Please find attached a revised version of our manuscript “Cutaneous melanoma: An overview of physiological and therapeutic aspects and biotechnological use of serine protease inhibitors”.

The comments of the reviewers were enlightening, improving the quality of our paper. In the next pages, are our present the changes suggested by the reviewers.

Comments and Suggestions for Authors

Boleti et al. wrote a review about cutaneous melanoma. The authors talked about the epidemiology and subtypes of melanoma. Then the authors discussed the staging phases and therapeutic strategies including the treatment with protease inhibitors. This will be a useful review for cutaneous melanoma. However, the current manuscript lacks figure illustration and needs more work on organization and typo errors. The following issues should also be resolved to improve the presentation of this review before consideration for publication.

 Comments:

  1. Lines 43-44: "ABCDE" signs were mentioned here. Only "A", "B", "C", and "D" were explained. Just curious, what does "E" stand for? Or "E" just make it rhyme.

Response:  The authors are in according with reviewer and in order to improve the information, the manuscript was revised.

Now reads, page 2, line 75-77:

“The initial stage, RGP, exhibits the ABCDE signs: "A" for asymmetry, "B" for irregular border, "C" for color variation, "D" for diameter (>4 mm initially), and "E" for evolution.”

  1. Line 94: “global incidence has increased, especially in age groups and men.” Did you mean a group of specific age here?

Response: The reviewer observation is absolutely right, the changes were realized to better clear the information.

Now reads, page 3, line 124-126:

“For all genders, there is a significant increase in incidence in the age group of 50 years or older. Although mortality rates and trends have declined in recent decades, the global incidence has risen, particularly among older age groups (over 50) and men [24].

  1. Multiple typos: line 102: “doing”, which organ did you want to mention?; line 131: “DSC”, I guess this should be “CSD”; Line 229: “bsuch as”; line 585: “IPs”: This should be protease inhibitors.

Response: The reviewer observation is absolutely right, the changes were realized and typos have been fixed.

4.Line 256-258: “low-CSD (Copy-Number-Signature-low)”: In lines 130-131, CSD was defined as cumulated solar damage. However, line 256 shows that CSD means Copy-Number-Signature-low. Do you mean low-CNS here? Line 258: Were “disguised” meant to be used here?

Response: The authors are in according with reviewer and in order to improve the information, the manuscript was revised and and the acronyms corrected.

Now reads, page 8, line 357:

“Melanomas can arise from congenital moles in either the junctional or dermal regions. Junctional component lesions, especially those with a radial growth pattern (RGP), may bear similarities to low-CNS (Copy-Number-Signature-low) melanomas.”

  1. Paragraph spacing is not consistent across the manuscript. e.g. lines 271-277 and lines 355-363.

Response: The authors are in agreement with reviewer and the spacing has been matched.

  1. The numbering of sections 3, 4, and 5 is not consistent and messed up. Each section should have its separate numbering and need to be consistent in the same section.

Response: The authors are in agreement with the reviewer and the section numbers have been checked and corrected.

  1. Section 5.2.1 and 5.2.2: What are the definitions of topical treatment and targeted therapy and how to perform them? Without a clear explanation, it seems that they are like regular chemotherapy.

Response: The reviewer's observation is absolutely correct, the definitions of definitions of topical treatment and targeted therapy have been added in the manuscript.

Now reads, page 14, topic 5.2.1, line 556:

“The topical dermal route is easy to administer, non-invasive, and minimizes systemic effects, thus enhancing patient compliance. Semisolid dosage forms, such as gels, are particularly advantageous due to their ability to adhere well to the application site, prolong the release of active ingredients, and provide excellent spreadability. Gels also tend to have fewer long-term stability issues compared to other semisolid forms and are relatively straightforward to formulate, accommodating a variety of active compounds [106]. According to Cancer Research UK, topical treatments are applied directly to the lesion site and are generally administered once or twice a day for 3 to 4 weeks, following medical advice [107].”

Now reads, page 14, topic 5.2.2, line 587:

“Targeted therapy is a cancer treatment that specifically targets proteins and genes responsible for regulating the growth, division, and spread of cancer cells. As our understanding of genetic mutations and protein mechanisms in cancer cells advances, researchers can develop more precise treatments [107]. In melanoma treatment, targeted therapy is particularly revolutionary because it causes less damage to healthy cells. These treatments focus on specific aspects of the immune system or on problematic genes and molecules, such as inhibiting the mitogen-activated protein kinase (MAPK) pathway in tumors with BRAF or MEK mutations [116], or targeting pathways like cyclin-dependent kinases, such as CDK4/6 [117].”

  1. Protease inhibitors can be small molecules, small proteins, or polypeptides. What are the protease inhibitors that have been studied so far? The authors mentioned a few small protein inhibitors in the paper. Are there any clinical trials using protease inhibitors? If the authors want to emphasize protease inhibitors here, more elaboration should be done. For example, a table summarizing the protease inhibitors with their types, effects, etc will be helpful.

Response: The authors are in according with reviewer’s suggestions, seeking to improve the exposure of the data obtained and the interpretation of this review, changes were realized throughout of the maintext and a table 2 was added.

Now reads, page 21, line 850:

“PIs are molecules that play a crucial role in regulating the activity of proteases, enzymes that catalyze the breakdown of peptide bonds in proteins. Excessive or dysregulated protease activity is associated with several diseases, including cancer, viral infections, inflammation, and neurodegenerative diseases [173]. Therefore, protease inhibitors have been widely studied for their potential therapeutic and biological effects, playing a critical role in therapies against various pathological conditions. Each class of protease inhibitors presents unique properties that can be harnessed to treat specific diseases, with ongoing research aimed at improving their efficacy and safety.

Serine protease inhibitors play a significant role in melanoma progression by modulation of signaling pathways and tissue remodeling [174]. Also, regulate the cell cycle, limit tumor growth, promote apoptosis, and reduce melanoma’s invasive and metastatic capacities. They impede the formation of new blood vessels that supply the tumor, which decreases the availability of essential nutrients for tumor growth [175]. Preclinical studies demonstrate significant efficacy in reducing the vision of melanoma cells in vitro and in vivo studies. However, clinical research is still in the early stages [166, 176]. In vivo assays are a critical step in investigating the therapeutic potential of new compounds, allowing the evaluation of efficacy, safety, and mechanism of action in a more complex biological context that more accurately mimics human physiology compared to in vitro assays. Table 2 shows the results of in vivo tests to evaluate the anti-melanoma potential of some protease inhibitors.”

  1. Most of this manuscript is about Melanoma subtypes, staging strategies, and treatments. Protease inhibitors were only discussed in section 6. However, the title, abstract, and future perspectives seem to emphasize the application of protease inhibitors in treating melanoma. Is there a particular reason to emphasize the protease inhibitors? The current manuscript does not show enough necessity or importance of protease inhibitors. For example, the use of protease inhibitors looks a lot like chemotherapy except that the protease inhibitors applied seem to be small proteins.

 Response: At this point authors partially agree with referee. Our review work focused on protease inhibitors because most authors researches these molecules. Our research laboratory (Protein Purification and Biological Functions Laboratory - LPPFB) has worked for over 30 years, isolating and characterizing proteins and peptides with biological functions, including cancer. Plant protease inhibitors are our source of research, patents, publications, and awards. We believe that Table 2 demonstrates the emphasis on the use of PI and its prospects in relation to cancer therapeutics.

10.It will be helpful if the authors can add more figure illustrations to ease the understanding of the main points of the manuscript. The current manuscript only has one figure. For a review paper, this is not enough to help readers understand the manuscript. E.g. figures showing different melanoma subtypes, figures illustrating topical treatments and targeted therapy, and figures showing protease inhibitor treatment and action of mechanism.

Response: Considering the suggestion a figure illustrating different types of treatment was added.

Comments on the Quality of English Language

A few typo errors were found in the manuscript. I mentioned a few in the comments. Please proofread the manuscript to solve the typos and grammar errors as needed.

Response: In order to improve grammatical and typographical errors in the article. The manuscript was revised, and errors were corrected

Thank you for your consideration of this manuscript.

Sincerely,

Profa. Dra. Maria Lígia Rodrigues Macedo

Laboratório de Purificação de Proteínas e Suas Funções Biológicas (LPPFB)

Universidade Federal de Mato Grosso do Sul

Postal Box 549, Campo Grande, Mato Grosso do Sul State, 79070-900, Brazil.

Round 2

Reviewer 2 Report

Comments and Suggestions for Authors

The revision is acceptable in the present form.

Comments on the Quality of English Language

Acceptable

Author Response

Editor Office

Molecules

Manuscript ID: molecules-3116056

Dear Reviewer:

Please find attached a revised version of our manuscript “Cutaneous melanoma: An overview of physiological and therapeutic aspects and biotechnological use of serine protease inhibitors”.

Thank you for contributing to the improvement of this manuscript.

Sincerely,

Profa. Dra. Maria Lígia Rodrigues Macedo

Laboratório de Purificação de Proteínas e Suas Funções Biológicas (LPPFB)

Universidade Federal de Mato Grosso do Sul

Postal Box 549, Campo Grande, Mato Grosso do Sul State, 79070-900, Brazil.

Reviewer 3 Report

Comments and Suggestions for Authors

I appreciate that the authors treat the reviewer’s comments seriously. The authors have addressed all my comments. The added figure 2 looks good and can help readers understand all the possible therapies for cancers. The new Table 2 is also useful and can help readers understand the importance and functions of protease inhibitors. More materials about protease inhibitors were also provided. Maybe the authors can also add cancer therapy using protease inhibitors to Figure 2. I also want to clarify comments 8 and 9. I didn’t mean that melanoma therapy using protease inhibitors is not necessary or important. I was trying to express an opinion that the information about protease inhibitors in the original manuscript is insufficient. I appreciate the over 30 years of work that the authors have done on functional proteins and peptides. I have no more concerns except the following two minor comments. 

Comments:

1.     Lines 261: “n addition,” should be “in addition”.

2.     The figures seem to have very low resolution.

Author Response

Editor Office

Molecules

Manuscript ID: molecules-3116056

Dear Reviewer:

Please find attached a revised version of our manuscript “Cutaneous melanoma: An overview of physiological and therapeutic aspects and biotechnological use of serine protease inhibitors”.

The reviewers' comments were informative and improved the quality of our article. Below are the changes suggested by the reviewer.

Reviewer comments:

Comments and Suggestions for Authors

I appreciate that the authors treat the reviewer’s comments seriously. The authors have addressed all my comments. The added figure 2 looks good and can help readers understand all the possible therapies for cancers. The new Table 2 is also useful and can help readers understand the importance and functions of protease inhibitors. More materials about protease inhibitors were also provided. Maybe the authors can also add cancer therapy using protease inhibitors to Figure 2. I also want to clarify comments 8 and 9. I didn’t mean that melanoma therapy using protease inhibitors is not necessary or important. I was trying to express an opinion that the information about protease inhibitors in the original manuscript is insufficient. I appreciate the over 30 years of work that the authors have done on functional proteins and peptides. I have no more concerns except the following two minor comments. 

Comments:

  1. Lines 261: “n addition,” should be “in addition”.

Response: Considering the suggestion, the spelling error has been corrected.

Now reads, page 6, line 261:

“In addition, acral melanoma frequently exhibits complex structural alterations, including amplifications and deletions of chromosomes, particularly affecting the cyclin D1 (CCND1), cyclin-dependent kinase 4 (CDK4), and telomerase reverse transcriptase (TERT) genes [56].”

  1. The figures seem to have very low resolution.

Response: The authors agree with the reviewer's suggestions, the resolution of the graphical abstract and figures has been improved. We have also included the treatment with protease inhibitors in figure 2.

 Thank you for your consideration of this manuscript.

Sincerely,

Profa. Dra. Maria Lígia Rodrigues Macedo

Laboratório de Purificação de Proteínas e Suas Funções Biológicas (LPPFB)

Universidade Federal de Mato Grosso do Sul

Postal Box 549, Campo Grande, Mato Grosso do Sul State, 79070-900, Brazil.
